

# Towards robust community assessments of the Earth's climate sensitivity

Kate Marvel[1] and Mark Webb[2]

[1]NASA Goddard Institute for Space Studies, New York, NY, USA
[2]Met Office Hadley Centre, Exeter, UK

**Correspondence:** Kate Marvel kate.marvel@nasa.gov

**Abstract.** The eventual planetary warming in response to elevated atmospheric carbon dioxide concentrations is not precisely known. This climate sensitivity $S$ depends primarily on the net physical climate feedbacks, usually denoted as $\lambda$. Multiple lines of evidence can constrain this feedback parameter: proxy-based and model evidence from past equilibrium climates, process-based understanding of the physics underlying changes, and recent observations of temperature change, top-of-atmosphere

energy imbalance, and ocean heat content. However, despite recent advances in combining these lines of evidence, the estimated range of $S$ remains large. Here, using a Bayesian framework, we discuss three sources of uncertainty: uncertainty in the evidence, structural uncertainty in the model used to interpret that evidence, and differing prior beliefs, and show how these affect the conclusions we may draw from a single line of evidence. We then propose a method to combine multiple estimates of the evidence, multiple multiple explanatory models, and the subjective assessments of different experts in order to arrive at

an assessment of $\lambda$ (and hence, climate sensitivity S) that may be rapidly updated as new information arrives and truly reflects the existing community of experts.

## 1 Introduction

When a radiative forcing $\Delta F$ is applied to the climate system, it induces a radiative imbalance $\Delta N$ at the top of the atmosphere

and a response $\Delta R$ of the system itself. To first order, $\Delta R = \lambda \Delta T$, where $\Delta T$ is the change in global mean surface temperature. The feedback parameter $\lambda$ thus measures the additional radiative flux density exported to space per unit warming. On sufficiently long timescales, internal variability is negligible and we can write a simple energy balance model for the climate system:

$$\mathcal{M}_0 : \Delta N = \Delta F + \lambda \Delta T. \tag{1}$$

In the special case where the radiative forcing results from a doubling of atmospheric $CO_2$ relative to its preindustrial concentration of 280ppm and the system is allowed to come into equilibrium ($\Delta N = 0$), then the internal variability term is negligible





and the resulting temperature change defines the equilbrium climate sensitivity $S$:

$$\Delta T_{2 \times CO2} \equiv S = -\frac{F_{2 \times CO2}}{\lambda}. \tag{2}$$

S is often used as a metric to quantify expected warming in response to radiative forcing, but has remained stubbornly uncertain
even as climate models have improved and become more sophisticated. A 2020 community assessment ((Sherwood et al., 2020), hereafter S20) reduced this range using multiple lines of evidence, but the recent IPCC report (Forster and Zhang, 2021) assessed only "medium confidence" in the upper bound. It is therefore imperative to reduce the uncertainty and enhance confidence in a quantity so crucial to climate science and policy.

S is determined by the net feedbacks $\lambda$ at equilibrium and in response to doubled $CO_2$. While these are unobservable in
the current system, in which $CO_2$ has not yet doubled and which is out of equilibrium, there exist several lines of evidence that might constrain $\lambda$. We have some process-based understanding of individual feedback processes and their correlations derived from observations and basic physics. We also have the evidence of the planet itself, which has been steadily warming in response to anthropogenic emissions of $CO_2$, other greenhouse gases, and aerosols. Finally, we have proxies that provide evidence about equilibrium climates of the past. S20 attempted to synthesize these three lines of evidence, incorporating the
judgement of many experts, and arrived at constraints on climate sensitivity that narrowed the former range.

This paper presents some lessons learned by two authors of S20 and attempts to chart a way forward. Our goals are to 1) better understand the sources of uncertainty 2) understand where unavoidable subjective decisions enter in to the analysis and 3) present a framework for systematically and fairly incorporating the subjective judgements of multiple experts.

The paper is organized as follows. In section 2, we review the basic Bayesian analysis framework. Sections 3, 4, and 5 discuss
evidence, structural, and prior uncertainty, respectively. In these sections, we use a single line of evidence– paleoclimate data from the Last Glacial Maximum– to illustrate how these sources of uncertainty shape estimates of climate feedbacks and sensitivity. In Section 6 we show how these sources of uncertainty affect constraints derived from multiple lines of evidence. In Section 7 we propose a new method for combining multiple published studies and multiple models, which may be used in the future to arrive at a robust community assessment of climate sensitivity. Finally, in section 8 we discuss possible generalizations
and extensions.

## 2 Analysis framework

As in S20, we use a Bayesian framework in which researchers express their beliefs about model parameters in terms of probability distributions. Bayesian statistics is both praised and criticized for its inherent subjectivity (see, e.g. (Gelman et al., 1995)). The framework requires researchers to specify prior distributions, which in general reflect some degree of knowledge
(Gelman et al., 2017). Different researchers may quite reasonably have different beliefs if they do not have access to the same observations or disagree with one another about the credibility of lines of evidence. The requirement to specify their priors forces researchers to be explicit about assumptions and pre-existing beliefs that might otherwise be implicit but ignored. After all, *every* statistical analysis contains subjective choices to some extent, whether is is the choice of the $p$-value threshold





in frequentist statistics or the selection of priors in Bayesian frameworks. Moreover, every statistical analysis depends on the
model used to interpret the evidence. Here, we show where unavoidable choices enter into the analysis framework, differentiate
between uncertainty in the model, uncertainty in the evidence, and uncertainty in the parameters, and suggest strategies to
handle and reduce these uncertainties in future work. We will introduce Bayesian hierarchical modeling approaches to weight
alternative models, evidence sources, and expert beliefs. Our hope is to lay the groundwork for future syntheses of evidence
and to constrain the use of "expert judgement" within a clearly defined, well-constrained framework.

Bayes' Theorem states that, given evidence $Y$ and a model with parameters $\Theta$, we can update our prior beliefs $P(\Theta)$ using

$$P(\Theta|Y) = \frac{P(Y|\Theta)P(\Theta)}{P(Y)}. \tag{3}$$

The term $P(Y|\Theta)$ is the likelihood, defined as the probability of the data given some putative values of the parameters
$\Theta$. The data $Y$ consist of multiple lines of evidence $Y_1 \ldots Y_n$; S20 used direct observations, models of varying complexity,
process-based understanding of underlying physics, and proxy-based reconstructions of past climates. In section 3 we will
discuss how differing interpretations of the evidence introduce uncertainties into the analysis.

The parameters $\Theta$ are specified by an underlying generative model $\mathcal{M}$ for the data. For example, the simplest energy balance
model $\mathcal{M}_0$ (Equation 1) contains a single parameter ($\Theta = \lambda$). This model carries with it the implicit assumptions that the
feedback parameter is constant in time, that a unit of global mean radiative forcing always produces the same temperature
change, and that the radiative imbalance $\Delta N$ always has unit efficacy. All of these assumptions have been challenged in the
literature (e.g.(Andrews et al., 2018; Winton et al., 2010; Forster, 2016; Dong et al., 2020; Rose et al., 2014; Armour et al.,
2013)). Other, more complex models with additional parameters are possible, and we will discuss these in Section 4.

Finally, the term $P(\Theta)$ corresponds to subjective prior beliefs about the values of the model parameters. While in theory
sufficient evidence should update the priors of all reasonable analysts and result in similar posterior estimates, in practice sparse
evidence and strongly held beliefs mean consensus may not be reached after the analysis is performed. In Section 5 we will
discuss how these subjective prior beliefs affect our estimates of the feedback parameter $\lambda$.

## 3  Evidence uncertainty

The first type of uncertainty we highlight is uncertainty in the *evidence*. Evidence-based constraints on climate sensitivity (S)
or feedback parameters ($\lambda$) are generally derived from estimates of temperature response, energy imbalance, and radiative
forcing. These may come directly from observations (e.g. the instrumental warming record or measured ocean heat uptake),
from model calculations which are informed by observations and theory (e.g. radiative forcing) or from observational evidence
which is interpreted using modeling assumptions.

In S20, the strongest constraints on the upper bound of warming were derived from paleoclimate evidence. In this section, we
will show how uncertainty in the evidence affects our confidence in those constraints. The closest equilibrium climate to ours
is the Last Glacial Maximum (LGM) approximately 21,000 years ago. Reconstructions (Annan and Hargreaves, 2013; Bereiter
et al., 2018; Friedrich et al., 2016; Holden et al., 2009; von Deimling et al., 2006; Shakun et al., 2012; Snyder, 2016) or model-



based estimates(Braconnot et al., 2012; Kageyama et al., 2021) of the global mean temperature change $\Delta T$ and the radiative forcing $\Delta F$ have been used to calculate the global mean feedbacks $\lambda$ inferred from this period. Neither of these "observed" quantities is precisely known. For example, multiple, seemingly incompatible, estimates of the LGM global mean cooling $\Delta T$ are available in the published literature (Annan and Hargreaves, 2013; Holden et al., 2009; Shakun et al., 2012; von Deimling

et al., 2006; Friedrich et al., 2016; Hansen et al., 2023; Annan et al., 2022; Bereiter et al., 2018; Snyder, 2016; Kageyama et al., 2021). These are derived from climate models participating in the Paleoclimate Model Intercomparison Project (PMIP (Kageyama et al., 2021)) and combinations of models and various proxies, and are often in conflict with one another.

We will illustrate the impact of this uncertainty by comparing the evidence used in two recent studies. S20 used expert judgement applied to a literature review to estimate $\Delta T = -5$K with a 95% confidence interval of (-3.0K, -7.0K). However, a

contemporaneous study using an updated temperature reconstruction (Tierney et al 2020 (Tierney et al., 2020), hereafter T20) estimated both colder (mean -6.1K) and less uncertain (with a 95 % highest posterior density interval of -6.5 to -5.7 K) values for LGM cooling.

The two studies S20 and T20 also differ in their estimates of the radiative forcing that led to this temperature change. Both agree that it was colder 21,000 years ago because a change in orbital forcing, while negligible in the global mean,

led to the development of large, reflective ice sheets in the northern hemisphere and lower levels of atmospheric greenhouse gases. The forcings associated with orbital changes (Kageyama et al., 2021) and $CO_2$ (Siegenthaler et al., 2005) are relatively well-constrained; the forcings from other well-mixed greenhouse gases (Loulergue et al., 2008) and ice sheets less so but still informed by proxy and model evidence (Section 8), and those from dust (Mahowald et al., 2006; Albani and Mahowald, 2019),, other aerosols, and vegetation (Köhler et al., 2010) highly uncertain. While S20 estimated total radiative forcing in the LGM

to be N(-8.43, 2) W m$^{-2}$, T20 use a best estimate of -6.8 W m$^{-2}$ with a 95% confidence interval of 9.6 to -5.2 W m$^{-2}$.

Some uncertainty in the climate feedback $\lambda$ or the climate sensitivity $S$ therefore derives from uncertainty in the evidence used to constrain those parameters. Here, we will define *evidence* uncertainty as uncertainty in the joint probability density of the evidence $Y$. This is defined as $Y_{LGM} = (\Delta T, \Delta F)$: the global cooling and radiative forcing during the LGM. Assuming the uncertainty in $\Delta T$ is independent of the uncertainty in $\Delta F$, the resulting joint probability density functions derived from

S20 and T20 are shown in Figure 1(a). The estimates used in S20 are Gaussian and have large uncertainties (black contours), while uncertainties in both temperature and radiative forcing are smaller in T20 (red contours). T20 also treats non-greenhouse gas forcing as non-Gaussian.

Using the simple energy balance model (Eq. 1) and setting $\Delta N = 0$, the forcing is proportional to the temperature change, with the only parameter the net feedback $\lambda$. This means that if we knew the temperature change and radiative forcing *exactly*,

we would know the feedback $\lambda$, and if we knew the forcing at doubled $CO_2$, we would then know the climate sensitivity exactly. However, we do not know the exact cooling and radiative forcing during the LGM, but rather a joint probability density $\mathcal{J}(Y)$ for both. The model $\mathcal{M}_0$ imposes the requirement that given a fixed value of the parameter $\lambda$, all pairs of $(\Delta T, \Delta F)$ lie on the line with slope $\lambda$. Integrating the joint probability density along that line results in the likelihood of the evidence given $\lambda$





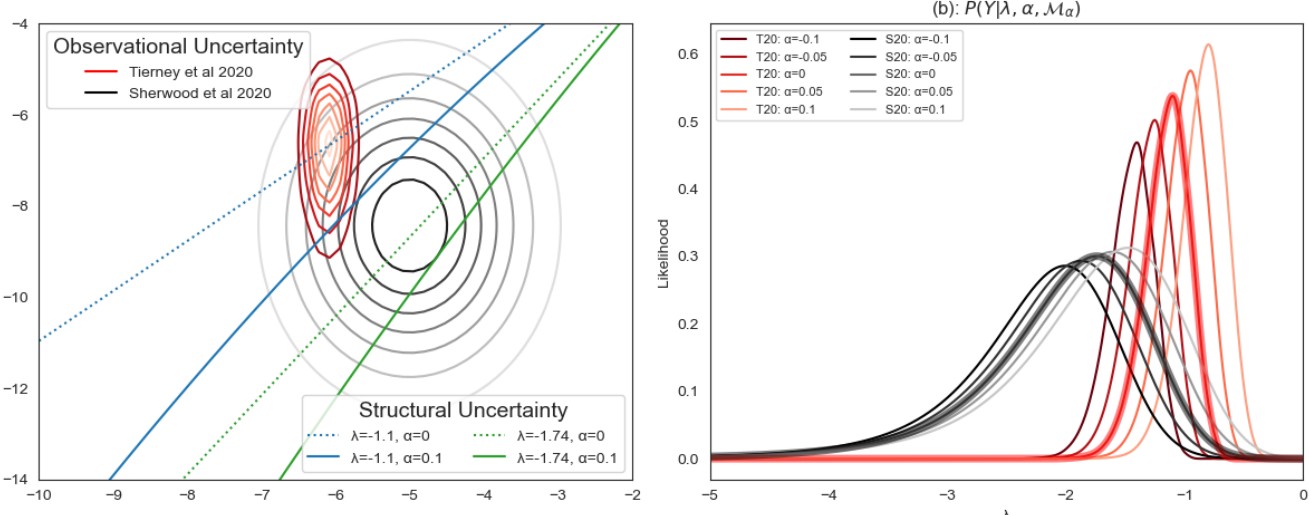

**Figure 1.** Panel (a): joint evidence distributions for $\Delta T$ and $\Delta F$ used in Sherwood et al (black contours) and Tierney et al (red contours). Structural uncertainty is illustrated using solid lines (corresponding to fixed values of $\lambda$ using the model $\mathcal{M}_0$) and dashed lines lines (corresponding to fixed values of $\lambda$ and $\alpha$ using the model $\mathcal{M}_\alpha$). (b): Likelihoods as a function of $\lambda$ and given S20 (black lines) or T20 (red lines) evidence and different values of the state dependence $\alpha$. (b): Resulting likelihoods for $\lambda$ given the evidence from S20 (black) and T20 (red) and different values of the state dependence parameter $\alpha$. Likelihoods derived using the simple energy balance model ($\alpha = 0$) are highlighted by thick lines.

(Figure 1(b)):

$$P(\Delta T, \Delta F|\lambda) \propto \int_C \mathcal{J}(\Delta T, \Delta F) ds$$
$$C : 0 = \Delta F + \lambda \Delta T$$

If the joint evidence is Gaussian, this leads to an exact analytic expression for $P(Y|\lambda)$ (Appendix 1). Using S20 evidence, this energy balance model, and a uniform prior $P(\lambda) = U(-10, 10)$ the most likely value of the feedback parameter is $\lambda = -1.7$ Wm$^{-2}$K$^{-1}$ (thick black line, Figure 1(b)) with a 5-95% range of (-3.37, -1.09) Wm$^{-2}$K$^{-1}$ . This corresponds to a 5-95%

range of (1.17K, 3.69K) for the climate sensitivity $S$ (assuming, as in S20, that $F_{2\times CO_2} \sim N(4.0, 0.3)$. Using T20 evidence, the most likely value is $\lambda = -1.1$ Wm$^{-2}$K$^{-1}$ ((thick red line, Figure 1(b)). The 5-95% range is (-1.49, -0.87) Wm$^{-2}$K$^{-1}$ for $\lambda$ and (2.61, 4.72) K for S. Clearly, the constraints placed on the climate feedback by the Last Glacial Maximum depend on our estimates of the temperature difference and the radiative forcing that caused it.

For simplicity, here we calculate the likelihood $P(\Delta T, \Delta F|\lambda)$, and use the resulting posterior $P(\lambda|\Delta T, \Delta F) \propto P(\Delta T, \Delta F|\lambda)P(\lambda)$

to calculate $S$ (Appendix 2). This neglects the small correlation between $\Delta F$ and the forcing at doubled $CO_2$, but this simplification does not substantially affect our results (Appendix 3).



## 4 Structural uncertainty

Many recent studies (e.g.(Rohling et al., 2018; Stap et al., 2019; Friedrich et al., 2016)) suggest that the simple model $\mathcal{M}_0$ might not be appropriate for past climates due to the dependence of the feedbacks on the background climate state. If the

relationship between temperature change and radiative forcing is nonlinear, then the feedbacks in a past cold climate should not be treated as identical to those in a future warm one. To model this background temperature dependence, we might use an alternate model that includes a second-order term in the radiative response

$$\mathcal{M}_\alpha : 0 = \Delta F + \lambda \Delta T + \frac{\alpha}{2} \Delta T^2 \qquad (4)$$

where $\alpha = \partial \lambda / \partial(\Delta T)$ is an additional parameter reflecting the background state dependence (Sellers, 1969; Caballero and

Huber, 2013; Budyko, 1969; Sherwood et al., 2020). Intuitively, nonzero values of $\alpha$ change the relationship between the paleoclimate evidence and the feedback parameter $\lambda$. This, in turn, makes the evidence more or less likely given a value of $\lambda$. For example, if $\alpha = +0.1$ (which translates to a change in feedback of -0.5 Wm$^{-2}$K$^{-1}$ at a cooling of -5 K), the most likely value of $\lambda$ is not the same as the most likely value of $\lambda$ assuming $\alpha = 0$ (dotted and solid lines, Figure 1a). In this case, the likelihoods (Figure 1b) are calculated by integrating the joint probability distribution for $\Delta T$ and $\Delta F$ along the curve defined

by Eq. 4, and depend on the value of the state dependence parameter $\alpha$.

If $\alpha$ is not a fixed value but an unknown parameter, then the evidence can constrain only the joint distribution of $\Theta = (\lambda, \alpha)$. Obviously, in order for the climate of the past to tell us anything about the climate of the future, we must have some information about how they relate to one another. There is no limit to the complexity of models we might use to interpret the evidence of the LGM. We might allow for both non-unit forcing efficacy and state dependence, for example, or assign different efficacies

to different forcing agents, or include an additive pattern effect $\Delta \lambda$ that reflects differences in the spatial pattern of temperature change in the LGM and the pattern of warming expected at elevated CO$_2$ concentrations.

The interpretive model is both required for analysis and subjectively chosen by the analyst. Different reasonable analysts might make different choices about the model to use. This means that the choice of model is an important source of uncertainty that must be clearly specified or quantified. There is, however, one more source of uncertainty to discuss. Even given a single

model, for example $\mathcal{M}_\alpha$, our degree of confidence in the constraints placed by paleoclimate evidence on the feedback parameter $\lambda$ necessarily reflects our prior knowledge of the state dependence of climate feedbacks. It is to this prior uncertainty that we turn in Section 5.

## 5 Prior uncertainty

Once a model is specified, we would like to use the evidence to tell us something about its parameters $\Theta$. Bayes' theorem

says that the posterior distributions- essentially, our degree of belief in the values of those parameters- are simply obtained by multiplying the likelihood by priors: probability distributions reflecting our pre-existing beliefs. These incorporate expert judgement, the results of other analyses, and knowledge of physical processes. Posterior distributions of individual parameters





can depend strongly on prior knowledge of all parameters. For example, Figure 2(a) shows the joint posteriors for the feedbacks
$\lambda$ and the state dependence $\alpha$ assuming the model $\mathcal{M}_\alpha$, the temperature and radiative forcing values reported in S20, and
uniform priors on both parameters. In the absence of any physical knowledge about these parameters, the joint posterior is not
very informative. In fact, considerable posterior weight is placed on extremely large positive values of $\alpha$ and positive $\lambda$: which
would make negative climate sensitivity appear more likely than most scientists would consider credible. A well-informed
scientist, however, is unlikely to think that $\alpha = 1$ (which implies an enormous mean change in feedback of -5 W m$^{-2}$ K$^{-1}$
for 5K of glacial cooling) is just as likely as $\alpha = 0$ (implying no change in feedback). In S20, a prior of $\mathcal{N}(+0.1, 0.1)$ was
assigned to the state dependence $\alpha$, reflecting the current state of the literature. This prior knowledge substantially constrains
the resulting joint posterior distribution (Figure 2 b). Conversely, imposing a more informative prior on the feedbacks $\lambda$, for
example using the process constraints in S20 that result in $\lambda \sim \mathcal{N}(-1.30, 0.44)$, also constrains the joint distribution: positive
values of $\alpha$ (i.e., which imply a lower sensitivity in the LGM than at doubled CO$_2$) receive more posterior weight. Combining
the informative priors on both $\lambda$ and $\alpha$ further constrains the joint posterior (Figure 2(d)).

Figure 3 summarizes the subjective decisions that must be made in any Bayesian analysis of climate feedbacks. First, the
analyst must decide what constitutes "evidence". In the LGM example we have presented, this requires subjective assessment
of the literature assesing the cooling $\Delta T$ and the radiative forcing $\Delta F$. Different highly credible published studies lead to very
different likelihoods for $\lambda$. Second, the analyst must specify a model (and its parameters $\Theta$) to interpret that evidence. For
example, the model $\mathcal{M}_0$ assumes the feedback parameter is time- and state-independent, and thus estimating it from the past
is a reliable guide to the hypothetical future under doubled CO2. The model $\mathcal{M}_\alpha$, by contrast, assumes the feedbacks depend
on the background climate and introduces a second parameter to reflect state dependence. Finally, the analyst must specify her
or his prior beliefs about the model parameters.

## 6  Combining multiple lines of evidence

The examples we have presented thus far have all used paleo evidence from the Last Glacial Maximum to constrain $\lambda$. More
recently, a large increase in radiative forcing has resulted in significant global warming and a large imbalance reflected in an
increased rate of ocean heat uptake. To constrain $\lambda$ with transient historical observations, the evidence $Y = (\Delta T, \Delta F, \Delta N)$
where $\Delta N$ is estimated from observed changes in ocean heat uptake and/or satellite observations constrained by ocean heat
content (Forster, 2016). In this three-dimensional joint probability space, the simplest energy balance model $\mathcal{M}_0$ defines a
plane rather than a line in evidence space (Figure 4), and the likelihood of the evidence given $\lambda$ is proportional to the integral
over this surface. Figure 4 shows the historical evidence reported in S20, in which

$$\Delta T \quad \sim \quad \mathcal{N}(1.03, 0.085) \tag{5}$$
$$\Delta N \quad \sim \quad \mathcal{N}(0.6, 0.18) \tag{6}$$



**Figure 2.** Joint posteriors for the feedback parameter $\lambda$ and the state dependence $\alpha$ under different priors: (a) Uniform priors on both parameters (b) Uniform prior on $\lambda$, Gaussian prior from expert judgement of published literature (used in S20) on $\alpha$ (c) Gaussian prior from process evidence (used in S20) on $\lambda$, uniform prior on $\alpha$ (d) Gaussian priors (from S20) on both.



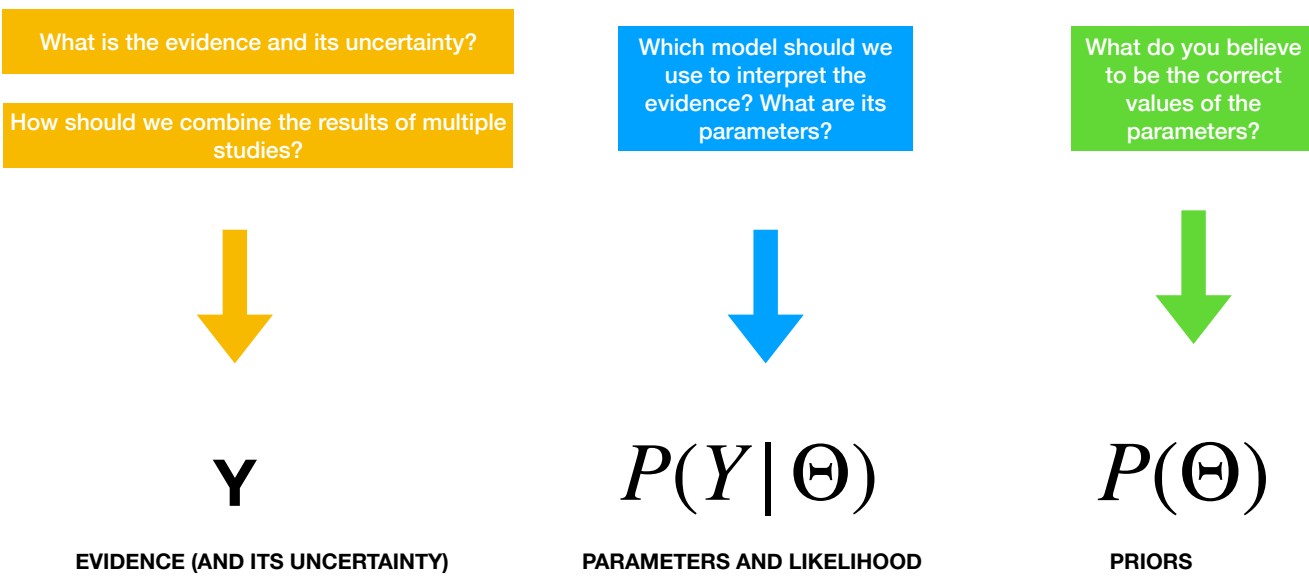

**Figure 3.** Schematic of unavoidable subjective decisions in an analysis of climate feedbacks.

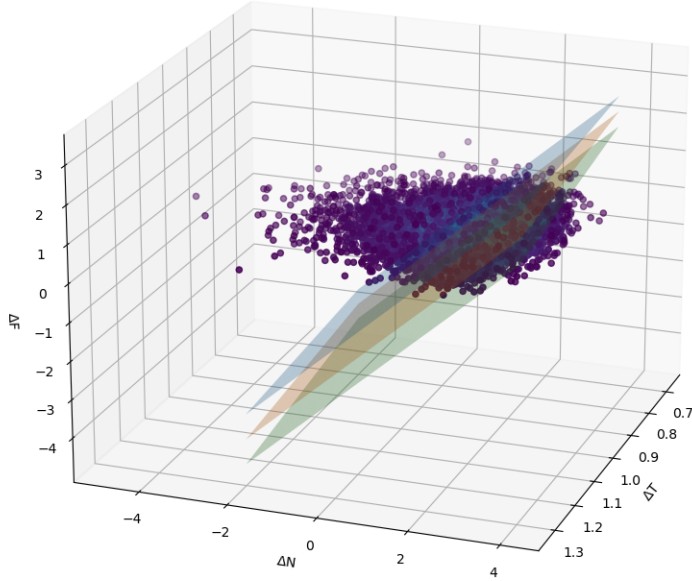

**Figure 4.** Calculating the likelihood of observing the historical evidence used in S20 for a putative value of $\lambda$. Each value of $\lambda$ defines a plane; shown are $\lambda = -1\mathrm{Wm}^{-2}\mathrm{K}$ (blue), $\lambda = -1.5\mathrm{Wm}^{-2}\mathrm{K}$ (orange) and $\lambda = -2\mathrm{Wm}^{-2}\mathrm{K}$ (green). The likelihood is the surface integral of the joint PDF along the plane.





and $\Delta F$ is calculated using unconstrained aerosol ERFs from (Bellouin et al., 2020) with median 1.83 W m$^{-2}$ and 5-95% range (-0.03, 2.71) W m$^{-2}$. The gray line in Figure 5 shows the resulting likelihood as a function of $\lambda$. The maximum likelihood

value is $\lambda = -1.53 \mathrm{Wm}^{-2}\mathrm{K}^{-1}$.

However, the simplest energy balance model $\mathcal{M}_0$ assumes the feedback parameter is the same for climate changes in the deep past, the transient historical period, and the future. Many studies (e.g. (Marvel et al., 2016; Andrews et al., 2018; Dong et al., 2020; Rose et al., 2014; Armour et al., 2013; Gregory and Andrews, 2016; Marvel et al., 2018)) now argue that a more appropriate model should include a "pattern effect" that reflects the differences between feedbacks triggered by the observed

pattern of transient warming and the feedbacks expected in response to the long-term equilibrium warming pattern:

$$\Delta\lambda = \frac{\partial\lambda}{\partial T'(x)}\Delta T'(x). \tag{7}$$

This modifies the simple energy balance model by including a pattern effect $\Delta\lambda$:

$$\mathcal{M}_{\Delta\lambda} : \Delta N = (\lambda - \Delta\lambda)\Delta T + \Delta F$$

S20 placed a Gaussian prior on this pattern effect $\Delta\lambda =$N(0.5 , 0.3) W m$^{-2}$K$^{-1}$. This corresponds to a modification of the tilt of the plane in Figure 4. Because this model assumes the pattern effect is linearly additive, no further curvature is introduced. By multiplying the joint likelihood $P(\Delta Q, \Delta T, \Delta F|\lambda, \Delta\lambda)$ by this prior $P(\Delta\lambda)$ and integrating over all values of $\Delta\lambda$, we

obtain a "marginal" likelihood for the historical evidence as a function of the feedback parameter $\lambda$. This is shown by the black line in Figure 5. The inclusion of the additive pattern effect and our prior belief that it is likely to be positive shift the most likely value of the feedback parameter to $\lambda = -1.0\mathrm{Wm}^{-2}\mathrm{K}^{-1}$.

### 6.1 Comparing "apples to apples" when combining lines of evidence

How should we ensure that when we combine multiple lines of evidence, we can be confident that these lines are actually

measuring the same thing? Bayesian methods of model evaluation provide useful checks to ensure we are indeed comparing "apples to apples".

Here is an illustrative example: suppose we firmly believe in the existence of a historical pattern effect, and we place Gaussian priors (the same as used in S20) on $\Delta\lambda$. These priors reflect our beliefs that this pattern effect acts to make future equilibrium feedbacks less negative (and thus climate sensitivity S higher).

How should we best compare these historical constraints with the evidence from the Last Glacial Maximum? The black line in all four panels of Figure 6 shows the likelihood for $\lambda$ derived from the historical observations (with a pattern effect). In Figure 6(a), the light blue line shows the likelihood for $\lambda$ derived from S20 LGM evidence, assuming no state dependence. Why do these two distributions not overlap substantially? If we are confident in the evidence used, the answer must be that the LGM and historical data are in fact measuring different things, and that additional steps must be taken to ensure that the distributions

reflect the same $\lambda$. The dark blue line in Figure 6(b) shows the marginal likelihood for $\lambda$ given the same LGM evidence, a model that allows for state dependence, and a Gaussian prior on the state dependence parameter $\alpha$. While the overlap between these two distributions is far from exact, it is substantially larger than for the no-state-dependence case illustrated in Figure





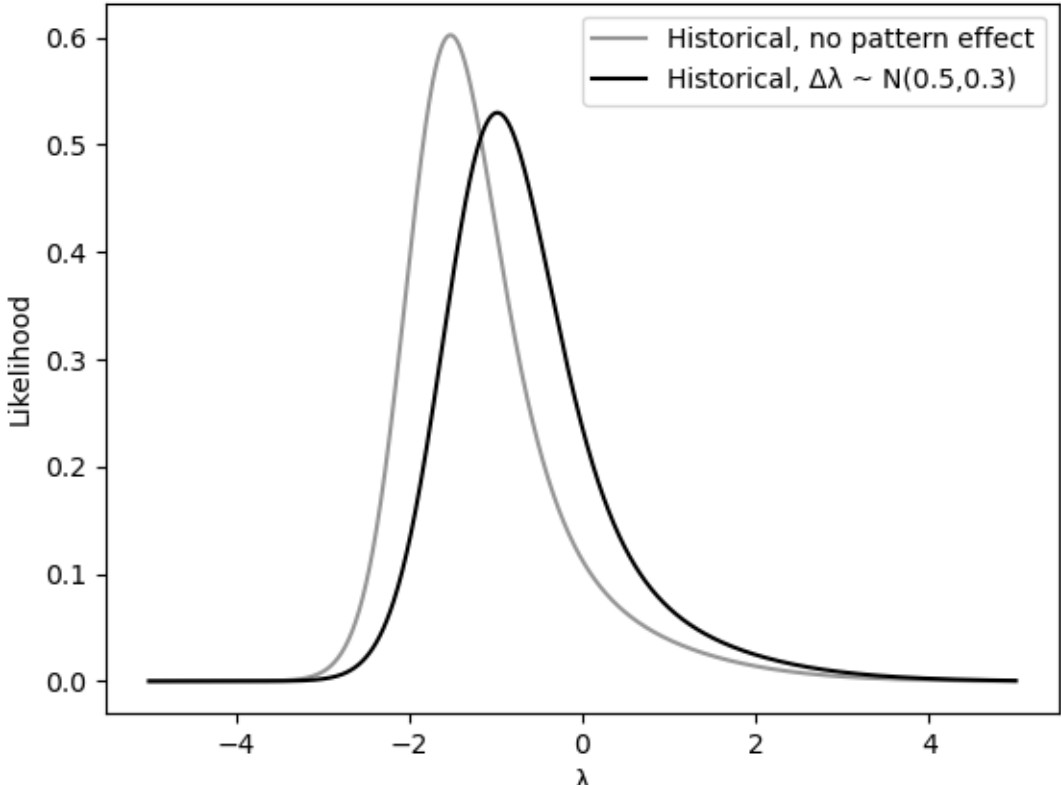

**Figure 5.** Likelihood for the feedback parameter $\lambda$ given the simple energy balance model with no pattern effect (gray line) and marginal likelihood for $\lambda$ given an additive pattern effect with prior $\Delta\lambda \sim N(0.5, 0.3)$.

6(a). Intuitively, the estimates appear to be in better agreement when Last Glacial Maximum estimates of $\lambda$ are assumed to be different from the historical estimates of $\lambda$.

The *model evidence* for any given model $\mathcal{M}_\ell$ can be calculated using

$$P(Y|\mathcal{M}_\ell) = \int P(Y|\Theta, \mathcal{M}_\ell) P(\Theta|\mathcal{M}_\ell) d\Theta. \tag{8}$$

This reflects the probability that model $\mathcal{M}_\ell$ could have generated the observed evidence under a given set of prior beliefs about its parameters $\theta$. The ratio of evidence for two models $\mathcal{M}_i$ and $\mathcal{M}_j$ defines the Bayes factor (BF), which updates the prior odds in favor or against a model. We note that the model evidence depends both on the model used and on the priors for

each parameter in the model. This has the useful property of penalizing models with many parameters that do not add value in fitting the evidence, thereby avoiding over-fitting.





**Figure 6.** Likelihoods from multiple lines of evidence. In all four panels, the black line shows the likelihood for the historical evidence given $\lambda$ and assuming a pattern effect $\Delta\lambda \sim N(0.5, 0.3)$. (a): Likelihood of S20 evidence given $\lambda$ assuming no state dependence in the LGM (light blue line) and overlap (dashed green line). (b):Likelihood of S20 evidence given $\lambda$ assuming state dependence and $\alpha \sim N(0.1, 0.1)$ (dark blue line) and overlap (dashed green line). (c): Likelihood of T20 evidence given $\lambda$ assuming no state dependence in the LGM (orange line) and overlap (dashed green line). (b):Likelihood of T20 evidence given $\lambda$ assuming state dependence and $\alpha \sim N(0.1, 0.1)$ (dark red line) and overlap (dashed green line).





Consider, for example, comparing the model used in Figure 6a to the model used in Figure 6b. In panel (a), the model assumes no state dependence for the LGM evidence but a pattern effect in the transient historical evidence; we will denote this as $\mathcal{M}_{0,\Delta\lambda}$. In panel (b), the model assumes the feedbacks depend on the background temperature and places the Gaussian prior used in S20 on the parameter $\alpha$, which we denote as $\mathcal{M}_{\alpha,\Delta\lambda}$. Using Eq. 8, the evidence for the first model $\mathcal{M}_{0,\Delta\lambda}$ is given by

$$P(Y_{hist}, Y_{paleo}|\mathcal{M}_{0,\Delta\lambda}) \propto \int\int P(Y_{paleo}|\lambda)P(Y_{hist}|\lambda,\Delta\lambda)P(\lambda)P(\Delta\lambda)d\lambda\,d(\Delta\lambda)$$

The terms in the integrand are, from left to right: the likelihood of the paleo evidence given $\lambda$, the likelihood of the historical evidence given $\lambda$ and $\Delta\lambda$, and priors on the feedbacks $\lambda$ (here, assumed uniform on -10,10) and the pattern effect $\Delta\lambda$ (as in S20, assumed to be N(0.5,0.3)).

The evidence for the second model is

$$P(Y_{hist}, Y_{paleo}|\mathcal{M}_{\alpha,\Delta\lambda}) \propto \int\int P(Y_{paleo}|\lambda,\alpha)P(Y_{hist}|\lambda,\Delta\lambda)P(\lambda)P(\Delta\lambda)P(\alpha)d\lambda\,d(\Delta\lambda)\,d\alpha$$

We can then calculate the Bayes factor

$$BF = \frac{P(Y|\mathcal{M}_{\alpha,\Delta\lambda})}{P(Y|\mathcal{M}_{0,\Delta\lambda})} = \frac{P(Y_{hist}, Y_{paleo}|\mathcal{M}_{\alpha,\Delta\lambda})}{P(Y_{hist}, Y_{paleo}|\mathcal{M}_{0,\Delta\lambda})} = 1.33.$$

If our prior belief is that both models are equally likely, the evidence shifts those odds: the model depicted in panel (b) is about 33% more likely to have generated the observed paleo and historical evidence.

Does this mean that the evidence indicates the definite existence of state dependence in the paleoclimate data? Certainly not. It simply means that

1. given the LGM and historical evidence used in S20

2. assuming the only candidate models to interpret the LGM evidence are those with and without state dependence and

3. using S20's Gaussian priors on $\alpha$

that this is are the model that maximizes the agreement between separate lines of evidence. We are *not* arguing that this is the objectively "correct" way to combine the Last Glacial Maximum reconstructions with historical observations.

In fact, whether or not we are comparing "apples to apples" depends very heavily on the evidence we use. As in the top two panels, the black lines in Figure 6(c) and (d) show the historical likelihood assuming a pattern effect with S20's Gaussian prior. The likelihood for $\lambda$ obtained from T20 evidence and assuming no state dependence (orange line, Figure 6(c)) closely overlaps the historical likelihood, as does the likelihood assuming state dependence with a prior on $\alpha$ as in S20 (red line, Figure 6(c)). The latter model, however, yields a broader likelihood for $\lambda$ and therefore the region of overlap with the historical evidence is smaller. The Bayes factor using T20 evidence is calculated as

$$BF = \frac{P(Y|\mathcal{M}_{\alpha,\Delta\lambda})}{P(Y|\mathcal{M}_{0,\Delta\lambda})} = 0.93.$$

This suggests that the "better" model to use, given T20 evidence, is one without state dependence.



## 7   A Way Forward

In the sections above, we have demonstrated that constraints on the feedback parameter $\lambda$ depend heavily on the evidence, the model used to interpret that evidence, and prior beliefs about the parameters. Moreover, our ability to compare different lines of evidence also depend on the evidence, models and priors we use to do so. It might initially appear that we are doomed to wallow in subjectivity, with no hope of arriving at credible, usable estimates of $\lambda$ or $S$. However, it is possible to move forward by relying on a community of experts, all of whom must be willing to clearly specify their prior beliefs and update their understandings in light of evidence.

Any expert assessment of $S$, $\lambda$, or indeed any other climate-relevant parameter (e.g. the Zero Emissions Commitment or the Transient Climate Response) should:

1. Easily update estimates as new information comes in

2. Compare "apples to apples" when combining lines of evidence.

3. Handle differing expert opinion in a fair and systematic way.

Here, we present a suggested way forward for expert assessment. Every analysis will require subjective decisions; we seek to both make these decision points explicit and allow for the fair aggregation of different expert choices.

### 7.1   Assessing evidence uncertainty

Bayesian methods are useful because they easily allow for hierarchical modeling, in which we can formulate sub-models to account for information on multiple levels, and easily propagate uncertainties. One of the most useful applications of hierarchical modeling is Bayesian meta-analysis(Smith et al., 1995), frequently used in fields as diverse as psychology(Gronau et al., 2021), medicine (Sutton and Abrams, 2001), and ecology(Koricheva et al., 2013). To combine multiple studies, we assume:

$$y_i \sim \mathcal{N}(\theta_i, \sigma_i) \tag{9}$$

$$\theta_i \sim \mathcal{N}(\mu, \tau) \tag{10}$$

$$\mu \sim g(.) \tag{11}$$

$$\tau \sim h(.) \tag{12}$$

Here $y_i$ and $\sigma_i$ are the reported mean and uncertainty of the evidence (i.e., $\Delta T$, $\Delta N$, or $\Delta F$) from study $i$. The reported mean $y_i$ is assumed to be distributed about a some true (latent) value $\theta_i$. This parameter depends on the study $i$, and can roughly be thought of as the value around which all subsequent repetitions of the work would be distributed. Each of these study means $\theta_i$ are then assumed to be drawn from a distribution with common mean $\mu$ and inter-study spread $\tau$. The priors $g(.)$ on $\mu$ and $h(.)$ on $\tau$ reflect our prior beliefs about both the true value of the parameter $\mu$ and the design of the studies. In a "fixed effects" meta-analysis the inter-study spread $\tau$ is assumed to be zero. This means that all reported estimates $y_i$ share a common mean, and any differences are simply due to sampling error. By contrast, in a "random effects" meta-analysis there are




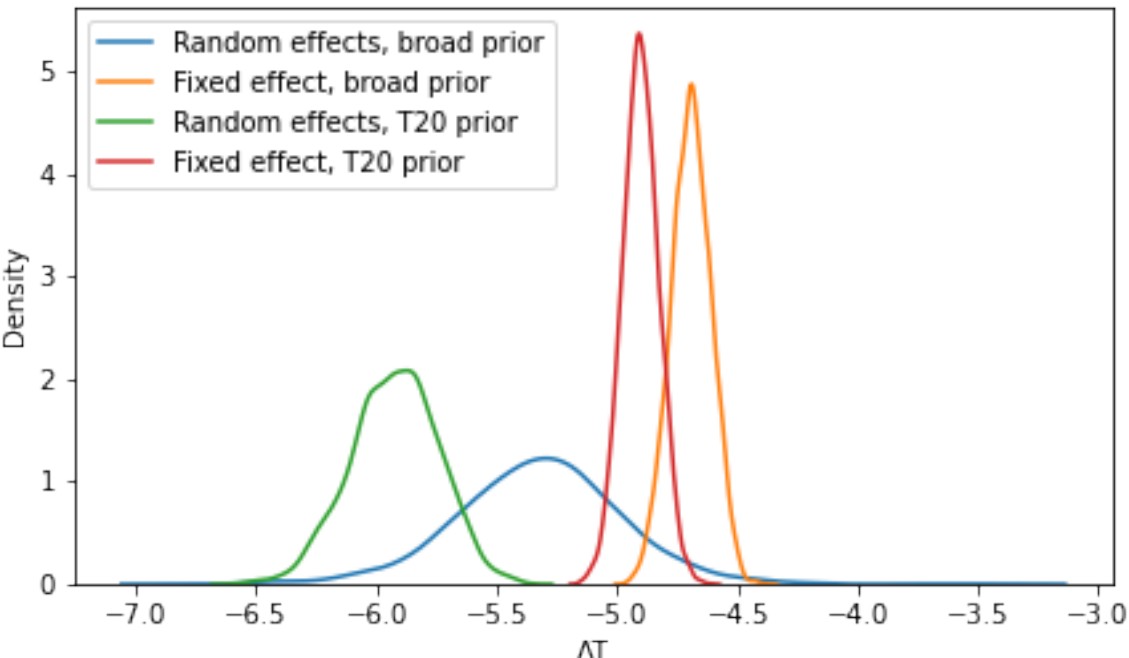

**Figure 7.** How cold was the Last Glacial Maximum? The answer depends on your prior beliefs about the cooling and about the literature. Shown are posterior distributions for the LGM cooling $\Delta T$ assuming a random effects model and broad (blue line) or T20 (green) priors on the mean or a fixed effects model and broad (orange line) or T20 (red line) priors on the mean.

assumed to be structural differences between individual studies that mean that we should expect variation between estimates. For example, a fixed-effects model might be appropriate for calculating the ensemble mean of a quantity within a single CMIP model, whereas a random-effects model might be more appropriate for combining ensembles of multiple CMIP models, which we know to differ structurally.

      As an specific example relevant to calculating the feedback parameter $\lambda$, we will consider multiple published LGM global

mean temperature changes $\Delta T$ derived from proxies and models as well as from PMIP3 and PMIP4 models (Table 1). How best should we combine these estimates to obtain a single distribution of $\Delta T$ and its uncertainty? This depends on many things: the literature on which we rely, our judgement about how best to pool multiple estimates into a single uncertain quantity, and the reported quantities in the studies themselves. Figure 7 illustrates how the posterior distribution of $\Delta T$ depends on prior beliefs about the nature and quality of the published literature assessing it. Consider, for example, a random-effects model in

which we place broad priors on the mean $\mu \sim N(0, 100)$ and inter-study standard deviation $\tau \sim U(0, 100)$. With these prior assumptions, 90% of the resulting posterior density for $\mu$ (the true value of $\Delta T$) lies between (-5.9K, -4.8K). Assuming that there is *no* inter-study spread (i.e, $\tau$ is assumed to be zero with zero uncertainty: a fixed effect model) would yield an estimate



| Mean (K) | Standard Deviation | Reference | Derived From | Generation |
|---|---|---|---|---|
| -4.00 | 0.41 | (Annan and Hargreaves, 2013) | Proxies and models | N/A |
| -5.80 | 0.77 | (von Deimling et al., 2006) | Proxies and models | N/A |
| -6.20 | 0.46 | (Holden et al., 2009) | GENIE-1 | N/A |
| -3.58 | 0.12 | (Shakun et al., 2012) | Proxies | N/A |
| -6.20 | 0.92 | (Snyder, 2016) | Proxies and models | N/A |
| -6.30 | 0.61 | (Bereiter et al., 2018) | Proxies (ocean temperature) and models | N/A |
| -5.70 | 0.20 | (Friedrich and Timmermann, 2020) | N/A | N/A |
| -5.75 | 0.38 | (Friedrich et al., 2016) | SST proxies and a model simulation | N/A |
| -6.10 | 0.20 | (Tierney et al., 2020) | proxies and isotope-enabled climate model | N/A |
| -5.00 | 1.00 | (Sherwood et al., 2020) | Synthesis | N/A |
| -4.85 | N/A | (Kageyama et al., 2021) | CESM | PMIP3 |
| -2.70 | N/A | (Kageyama et al., 2021) | CNRM | PMIP3 |
| -4.63 | N/A | (Kageyama et al., 2021) | FGOALS-g2 | PMIP3 |
| -4.92 | N/A | (Kageyama et al., 2021) | GISSE2-p1 | PMIP3 |
| -5.19 | N/A | (Kageyama et al., 2021) | GISSE2-p2 | PMIP3 |
| -4.64 | N/A | (Kageyama et al., 2021) | IPSL | PMIP3 |
| -5.40 | N/A | (Kageyama et al., 2021) | MIROC | PMIP3 |
| -4.41 | N/A | (Kageyama et al., 2021) | MPI-p1 | PMIP3 |
| -4.67 | N/A | (Kageyama et al., 2021) | MPI-p2 | PMIP3 |
| -4.71 | N/A | (Kageyama et al., 2021) | MRI | PMIP3 |
| -3.75 | N/A | (Kageyama et al., 2021) | AWIESM1 | PMIP4 |
| -3.81 | N/A | (Kageyama et al., 2021) | AWIESM2 | PMIP4 |
| -6.80 | N/A | (Kageyama et al., 2021) | CESM1-2 | PMIP4 |
| -7.16 | N/A | (Kageyama et al., 2021) | HadCM3-PMIP3 | PMIP4 |
| -5.92 | N/A | (Kageyama et al., 2021) | HadCM3-ICE6GC | PMIP4 |
| -6.46 | N/A | (Kageyama et al., 2021) | HadCM3-GLAC1D | PMIP4 |
| -3.28 | N/A | (Kageyama et al., 2021) | iLOVECLIM-ICE-6G | PMIP4 |
| -3.26 | N/A | (Kageyama et al., 2021) | iLOVECLIM-GLAC1D | PMIP4 |
| -3.73 | N/A | (Kageyama et al., 2021) | INM-CM4-8 | PMIP4 |
| -4.63 | N/A | (Kageyama et al., 2021) | IPSLCM5A2 | PMIP4 |
| -4.02 | N/A | (Kageyama et al., 2021) | MIROC-ES2L | PMIP4 |
| -3.90 | N/A | (Kageyama et al., 2021) | MPI-PMIP4 | PMIP4 |
| -5.27 | N/A | (Kageyama et al., 2021) | UT-CCSM4 | PMIP4 |

**Table 1.** Estimates of global cooling $\Delta T$ during the Last Glacial Maximum



of $\Delta T$ 90% likely to be between -4.8 and -4.5K. This much narrower estimate results from the extremely restrictive prior belief that every study, regardless of method, targets the same underlying $\Delta T$ and would yield the same results if performed perfectly

and with adequate data. Similarly, if we believe the result of a single published study, for example T20, to be accurate, then we may adopt this as our prior belief, setting the prior on $\mu$ to be the T20 distribution of $\Delta T$. Combined with a broad uniform prior on the inter-study spread, this results in an 90% posterior density estimate of (-6.2K, -5.6K). If, however, we adopt the restrictive fixed effects model, the T20 study is merely treated as an outlier and fails to substantially move the posterior distribution toward cooler values of $\Delta T$ (red line), even if our prior belief is that T20 is exactly correct.

What this means is that a simple hierarchical model for the evidence ($\Delta T$ or $\Delta F$, for example) allows different experts to specify their prior beliefs about the true values of the evidence and the literature. Their resulting posteriors will depend strongly on those prior assessments. This simply reflects the fact that different experts give different weights to studies in the published literature. It is a mathematical expression of the subjective expert judgment inherent in science. We propose a method for aggregating expert judgement in section 7.3.

To obtain new constraints on $\lambda$ from a meta-analysis of published LGM temperature estimates, we adopt broad priors on $\mu$ and $\tau$ (blue line, Figure 7). We also perform a similar meta-analysis for the radiative forcing using the values in Table 2 (assuming a fixed-effects model ($\tau = 0$) in the absence of reported standard deviations and broad priors N(0,100) on all means), which results in an estimate of $\Delta F = N(-8.1, 1.5)$ Wm$^{-2}$. Using these constraints and assuming no state dependence, the most likely value is $\lambda = -1.54$ with 5-95% range of (-2.09, -1.08)Wm$^{-2}$K$^{-1}$ , corresponding to a 5-95% range of (1.87,

3.75)K for S. We will explore the impact of different models and priors on this estimate in sections 7.2 and 7.3, respectively.

## 7.2   Handling model uncertainty

As shown in Section 4, the constraints placed on climate sensitivity by paleo or historical evidence depend on the model used to interpret that evidence. This means that the design of every expert assessment must be explicit about the models used to interpret each line of evidence. As the assessment is planned, it is crucial to arrive at consensus on credible interpretive

models for the evidence. For example, one possible model for the Last Glacial Maximum might incorporate parameters $\alpha$ (representing state dependence), $\xi$ (representing the difference between long-term equilibrium LGM feedbacks and the target quasi-equilibrium feedbacks to doubled CO2) and $\Delta\lambda_{LGM}$ (representing radiatively important sea-surface pattern differences between the LGM and doubled CO2):

$$\Delta T = \frac{-\Delta F}{\frac{\lambda + \Delta\lambda_{LGM}}{1+\xi} + \frac{\alpha}{2}\Delta T}$$

Given a model, even an unwieldy one with multiple parameters, experts may then be asked to specify their prior beliefs about each parameter. If an expert disagrees with the inclusion of a parameter in a model, s/he would be free to set a prior very narrowly clustered around 0 on that prior.



| Quantity | Mean (Wm$^{-2}$) | Standard Deviation | Reference | Model | Generation |
|---|---|---|---|---|---|
| $\Delta F_{ice}$ | -3.79 | N/A | (Braconnot and Kageyama, 2015) | CCSM4 | PMIP3 |
| $\Delta F_{ice}$ | -4.90 | N/A | (Braconnot and Kageyama, 2015) | IPSL-CM5A-LR | PMIP3 |
| $\Delta F_{ice}$ | -5.20 | N/A | (Braconnot and Kageyama, 2015) | MIROC-ESM | PMIP3 |
| $\Delta F_{ice}$ | -4.57 | N/A | (Braconnot and Kageyama, 2015) | MPI-ESM-P | PMIP3 |
| $\Delta F_{ice}$ | -3.62 | N/A | (Braconnot and Kageyama, 2015) | MRI-CGCM3 | PMIP3 |
| $\Delta F_{ice}$ | -2.59 | N/A | (Braconnot et al., 2012) | CCSM3 | PMIP4 |
| $\Delta F_{ice}$ | -2.66 | N/A | (Braconnot et al., 2012) | CMRM | PMIP4 |
| $\Delta F_{ice}$ | -3.23 | N/A | (Braconnot et al., 2012) | HadCM3M2 | PMIP4 |
| $\Delta F_{ice}$ | -3.41 | N/A | (Braconnot et al., 2012) | HadCM3M2 v | PMIP4 |
| $\Delta F_{ice}$ | -3.48 | N/A | (Braconnot et al., 2012) | IPSL-CM4 | PMIP4 |
| $\Delta F_{ice}$ | -2.88 | N/A | (Braconnot et al., 2012) | MIROC3.2 | PMIP4 |
| $\Delta F_{ice}$ | -3.29 | N/A | (Tierney et al., 2020) | CESM1.2 | PMIP4 |
| $\Delta F_{GHG}$ | -2.60 | 0.10 | (Zhu and Poulsen, 2021) | N/A | N/A |
| $\Delta F_{GHG}$ | -2.48 | 0.15 | (Tierney et al., 2020) | N/A | N/A |
| $\Delta F_{GHG}$ | -3.15 | 0.26 | (Sherwood et al., 2020) | N/A | N/A |
| $\Delta F_{Dust}$ | -0.12 | N/A | (Albani et al., 2014); (Albani and Mahowald, 2019) | C4fn-lgm | N/A |
| $\Delta F_{Dust}$ | -0.36 | N/A | (Hopcroft et al., 2015) | HadGEM2-A fixPIveg | N/A |
| $\Delta F_{Dust}$ | -1.10 | N/A | (Hopcroft et al., 2015) | HadGEM2-A | N/A |
| $\Delta F_{Dust}$ | -0.32 | N/A | (Hopcroft et al., 2015) | HadGEM2-A-DEAD | N/A |
| $\Delta F_{Dust}$ | -2.00 | N/A | (Claquin et al., 2003) | Exp1 (ext. mixing) | N/A |
| $\Delta F_{Dust}$ | -1.00 | N/A | (Claquin et al., 2003) | Exp2 (int. mix. Hem.) | N/A |
| $\Delta F_{Dust}$ | -0.48 | N/A | (Mahowald et al., 2006) | SOMB / SOMBLGMT | N/A |
| $\Delta F_{Dust}$ | -0.01 | N/A | (Takemura et al., 2009) | N/A | N/A |
| $\Delta F_{Dust}$ | 0.10 | N/A | (Yue et al., 2011) | PRND / LGM.DST | N/A |
| $\Delta F_{insolation}$ | 0.01 | N/A | (Braconnot and Kageyama, 2015) | CCSM4 | N/A |
| $\Delta F_{insolation}$ | 0.01 | N/A | (Braconnot and Kageyama, 2015) | IPSL-CM5A-LR | N/A |
| $\Delta F_{insolation}$ | 0.13 | N/A | (Braconnot and Kageyama, 2015) | MIROC-ESM | N/A |
| $\Delta F_{insolation}$ | 0.01 | N/A | (Braconnot and Kageyama, 2015) | MPI-ESM-P | N/A |
| $\Delta F_{insolation}$ | 0.01 | N/A | (Braconnot and Kageyama, 2015) | MRI-CGCM3 | N/A |
| $\Delta F_{vegetation}$ | -1.1 | 0.6 | (Köhler et al., 2010) | N/A | N/A |

**Table 2.** Estimates of $\Delta F$ for the Last Glacial Maximum from ice sheets, solar insolation, dust, and vegetation.





If consensus cannot be reached on a particular model, then we suggest that the planning team for any assessment arrive at a list of candidate models $\mathcal{M}_1 \ldots \mathcal{M}_K$. The aggregate posterior can then be taken as a weighted average over different models:

$$P(\Theta|Y) = \sum_{k=1}^{K} w_k P(\Theta|\mathcal{M}_k, Y). \tag{13}$$

Here, $(\Theta|M_k, Y)$ is the posterior obtained using the model $\mathcal{M}_k$ to interpret the evidence $Y$.

The weights reflect how well the model fits the data, and are given by

$$w_k = P(\mathcal{M}_k|Y) = \frac{P(Y|\mathcal{M}_k)P(\mathcal{M}_k)}{\sum_{k=1}^{K} P(Y|\mathcal{M}_k)P(\mathcal{M}_k)}. \tag{14}$$

The term $P(\mathcal{M}_k|Y)$ is the model evidence (Eq 8, discussed in section 6.1). These weights, and hence the combined posterior, depend on the priors $P(\mathcal{M}_k)$ we put on the correctness of each model. If an assessment allows for experts to use one of multiple models, it is therefore imperative to specify assessment-wide priors on these models upfront.

As a worked example, consider two models: $\mathcal{M}_{0,\Delta\lambda}$ in which the paleoclimate evidence is assumed to have no state dependence but a pattern effect is present in the historical observations, and $\mathcal{M}_{\alpha,0}$ in which there is no pattern effect in the historical observations but we allow for state dependence in the LGM. Table **??** shows the resulting estimates given different prior beliefs about these models. In all cases, the prior on $\lambda$ is assumed to be U(-10,10) and the prior on all parameters is as given in S20.

| Model Priors | $P(\Theta)$ | $\lambda(95\%CL)$ | $S(95\%CL)$ |
|:---:|:---:|:---:|:---:|
| $P(\mathcal{M}_{0,\Delta\lambda}) = 1$ | $P(\Delta\lambda) = N(0.5, 0.3)$ | (-1.9,-1.1) | (2.1,4.0) |
| $P(\mathcal{M}_{\alpha,0}) = 1$ | $P(\alpha) = N(0.1, 0.1)$ | (-1.9,-0.7) | (2.1,5.4) |
| $P(\mathcal{M}_{0,\Delta\lambda}) = P(\mathcal{M}_{\alpha,0}) = 0.5$ | as above | (-1.9,-0.8) | (2.1,5.0) |

**Table 3.** Estimates of feedbacks and climate sensitivity given different prior assumptions about models.

### 7.3 Expert elicitation via priors

Finally, it is necessary to quantify the degree of pre-existing knowledge and/or beliefs through the use of prior distributions. These enter the present analysis in three different places: first, in beliefs about the multiple studies used to constrain the evidence (priors on $\mu$ and $\tau$ in Section 7.1), second in beliefs about the underlying model used to explain the evidence and finally, in beliefs about the distributions of the parameters $\theta$ in those models.

In theory, sufficient evidence should lead to a high degree of agreement among experts, even if they begin the analysis with different prior beliefs. Figure 8(a) shows the prior beliefs of two hypothetical experts. Expert A (solid red line) believes the feedback parameter to be less negative than Expert B (solid blue line) and is even open to the idea that it might be positive. Dashed red and blue lines show both experts' posteriors, when updated using the evidence presented in S20. While the experts began their analysis with differing opinions, the weight of the evidence has updated their understandings and they now agree about the feedback parameter $\lambda$. However, some experts may not be as open-minded as our researchers A and B. Expert C





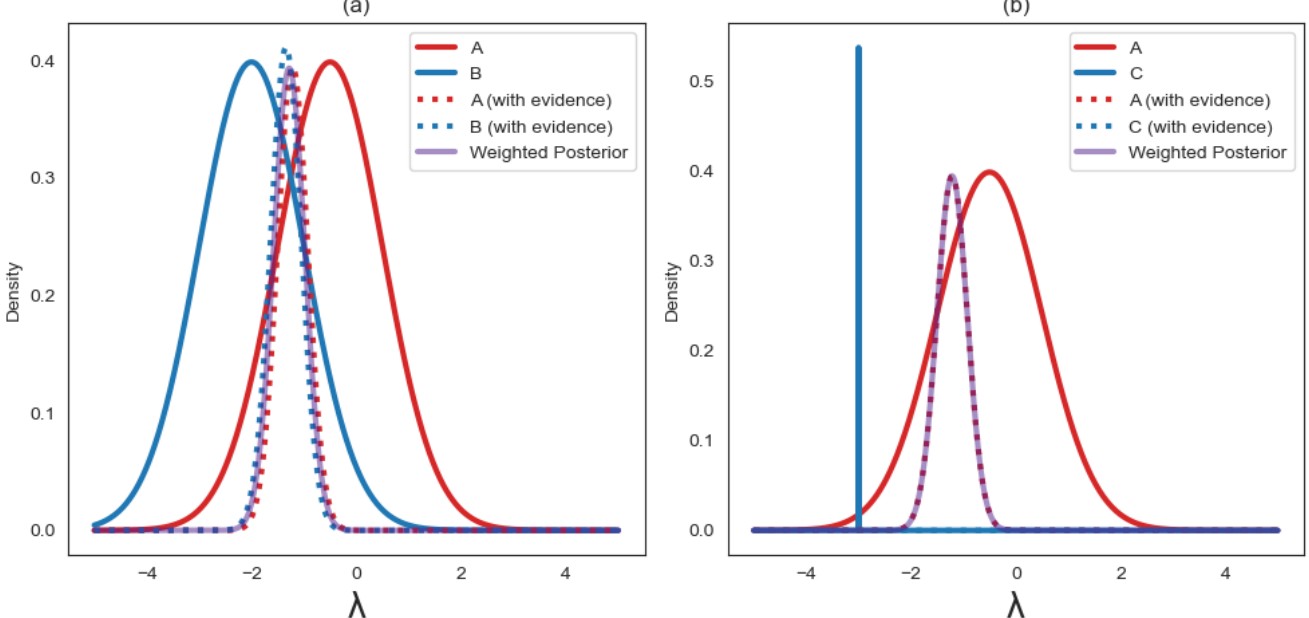

**Figure 8.** (a): Experts A (solid red line) and B (solid blue line) begin with different priors on $\lambda$. The evidence presented in S20 updates these priors, and the resulting posteriors are nearly identical (dotted red and blue lines). The purple line shows the weighted posterior. (a): Experts A (solid red line) and C (solid blue line) begin with different priors on $\lambda$, but C's prior is very narrowly peaked. The evidence presented in S20 updates these priors, but the posteriors remain very different(dotted red and blue lines). The purple line shows the weighted posterior, which is almost identical to A's posterior.

(blue line, Figure 8(b) believes the feedback parameter to be strongly negative. Moreover, he is extremely confident in this: his prior distributions is very narrowly peaked around a value of $\lambda = -3Wm^{-2}K^{-1}$. Expert C's confidence remains unshaken by the evidence presented in S20, and his posterior remains nearly identical to his prior beliefs. How should an assessment handle

345 such excessively confident experts, whose beliefs appear to be unshakeable by any reasonable amount of evidence?

Consider an assessment in which $N$ experts each specify their priors $P_i(\theta)$, where $i = 1 \ldots N$. A reasonable aggregate prior might then be a linear combination of the individual expert priors:

$$P(\theta) = \sum_{i=1}^{N} a_i P_i(\theta).$$

The aggregate posterior is therefore a weighted average of the individual expert posteriors

$$P(\theta, Y) = \sum_{i} \tilde{a}_i P_i(\theta | i, Y)$$

where

$$\tilde{a}_i = \frac{a_i \int P(Y|\theta) P_i(\theta) d\theta}{\sum_{i=1}^{N} a_i \int P(Y|\theta) P_i(\theta) d\theta}.$$





This method introduces $N$ new parameters: the prior weight $a_i$ we assign to each expert's judgement. This is a far easier task than setting priors on models (as discussed in Section 7.2) because it requires no physical understanding, only a belief about the "quality" of each expert's initial beliefs. We recommend weighting each expert equally by setting $a_1 = a_2 = \ldots a_N = \frac{1}{N}$, in which case the posterior weights become

$$\tilde{a}_i = \frac{\int P(Y|\theta)P_i(\theta)d\theta}{\sum_{i=1}^{N} \int P(Y|\theta)P_i(\theta)d\theta}.$$

The purple line in Figure 8(a) shows the resulting aggregate posterior given A and B's priors. Because both these experts are similarly able to update their priors, the weighting process has no effect on the outcome. However, the weighted average of A and C's posteriors, shown as a purple line in 8(b), is similar to A's posterior distribution. The narrowness of C's prior causes his posterior distribution to be down-weighted in the weighted average. We suggest this as an effective strategy for handling inflexible or extremely anomalous expert opinions.

## 8 Discussion and Conclusions

Here, we have presented three sources of uncertainty that enter in to estimates of climate sensitivity. First, what evidence are we using to constrain climate sensitivity, where do those estimates or measurements come from, and how should we handle estimates that disagree or conflict? Second, what interpretive model should we be using to relate the evidence to the climate sensitivity, and what parameters are required? Third, what prior knowledge of these parameters is it appropriate to include? In the subsequent section, we have laid out the rudiments of a strategy to combine multiple published estimates of variables relevant to climate sensitivity. The advantage of this strategy, combining Bayesian meta-analysis and Bayesian model averaging, is that it can incorporate newly published data and is easily expanded to handle uncertainties at multiple levels.

There is no limit to the number of nested levels we could theoretically use within a Bayesian hierarchical model: the prior for radiative forcing from ice sheets, for example, can be updated using a global ice sheet reconstruction, which itself is constrained by individual geological measurements. Similarly, a prior on ocean heat uptake $\Delta N$ or historical warming $\Delta T$ can be updated as new measurements become available. However, to remain tractable every project must truncate the hierarchy at some finite level. In practice, this means treating the posteriors that arise from observational, GCM, or paleoclimate studies as evidence; where we draw the line between evidence and parameter sets the bounds of our analysis.

As a result, we propose a framework in which experts are required to specify their choices at clearly defined decision points. Once priors are specified, the model and evidence will update them accordingly, arriving at a new, aggregate consensus posterior. We review this framework here.

Somewhat obviously, experts' beliefs about the data are based on their prior beliefs, updated by the evidence. But how they interpret and use that evidence depends on the subjective choices they make: what counts as a "study" or "evidence"? How should we best compare estimates derived from proxies or observations and estimates from GCMs? Should some studies receive more weight than others? In our framework, experts must make make the following judements about the evidence:

1. What is your informed belief about the evidence? (E.g. what is your prior on $\mu$?)





2. What is your belief about the published literature? (What is your prior on $\tau$)

Second, we suggest taking the choice of model out of individual participants' hands to the greatest extent possible. Ideally,
assessment planners would arrive at a single model and set of parameters on which experts may specify their priors. If not, they should arrive at a list of candidate models, specify firm prior beliefs about these models, and perform Bayesian model averaging over the posteriors of individual experts, which will depend on the model they use.

Third, once a model is specified, experts should specify their prior beliefs about the parameters of that model.

The results presented here are meant to begin, not end, a conversation. The beauty of Bayesian methods is that we can allow
new evidence to update our existing beliefs. As climate researchers gear up for the next generation of model intercomparison projects and assessments, it is important to consider how these new results will be integrated with existing knowledge. Our methods presented here allow for new discoveries to advance our understanding, ultimately narrowing the bounds of climate sensitivity and informing future research and decision making.

*Code availability.* All code is available at https://github.com/netzeroasap/LambdaBayes

*Data availability.* Data for Last Glacial Maximum forcing and temperature used in the meta-analysis are available at
     https://docs.google.com/spreadsheets/d/1jtNR-dxL3OHj8TQoU10P4PVMWpMDoVMODJMFNmk7R4Y/edit#gid=0

**Appendix A: 1**

**A1    Exact forms of integrals**

To estimate the likelihood of the evidence $\Delta T$ and $\Delta F$ given the simple energy balance model, we integrate the joint probability
distribution $\mathcal{J}(\Delta T, \Delta F)$ over the curve $C$ defined by the model :

$$P(Y|\lambda, \mathcal{M}_0) = \int_C \mathcal{J}(\Delta T, \Delta F) ds \tag{A1}$$

The curve $C$ can be parameterized as

$$\mathbf{r}(t) = t\hat{i} + -\lambda t \hat{j}$$

and the integral is then

$$P(Y|\lambda, \mathcal{M}_0) = \int_{-\infty}^{\infty} \mathcal{J}(\mathbf{r}(t)) \|\mathbf{r}'(t)\| \, dt = \int_{-\infty}^{\infty} \mathcal{J}(t, -\lambda t)\sqrt{1 + \lambda^2} \, dt. \tag{A2}$$





In the case where $\Delta T$ and $\Delta F$ are Gaussian and independent with means $\mu_T, \mu_F$ and standard deviations $\sigma_T, \sigma_F$ respectively,

the likelihood has an exact analytic form, substantially speeding up its computation:

$$P(Y|\lambda, \mathcal{M}_0) = C \left(\frac{2\pi}{A}\right)^{1/2} \exp\left(\frac{B^2}{2A}\right) \tag{A3}$$

where

$$
\begin{aligned}
C &= \frac{\sqrt{1+\lambda^2}}{2\pi\sigma_T\sigma_F} \exp\left(\frac{\mu_T^2}{\sigma_T^2} + \frac{\mu_F^2}{\sigma_F^2}\right) \\
A &= \frac{1}{\sigma_T^2} + \frac{\lambda^2}{\sigma_F^2} \\
B &= \frac{\mu_T}{\sigma_T^2} - \frac{\lambda\mu_F}{\sigma_F^2}
\end{aligned}
$$


In the case of a three-dimensional space (as for the historical evidence), the curve $C$ defines a plane, not a line, and we have

$$P(Y|\lambda) \propto \int_C \mathcal{J}(\Delta T, \Delta F, \Delta N)dS = \int\int \mathcal{J}\left(\boldsymbol{r}(u,v)\right)||r_u \times r_v||du\,dv$$

where

$$\boldsymbol{r} = u\hat{i} + v\hat{j} + (\lambda u + v)\hat{k}$$

In the case where the uncertainties in temperature, energy imbalance, and historical forcing are all Gaussian and uncorrelated, the likelihood can be evaluated analytically:

$$\mathcal{L}(\lambda_{hist}|\Delta T, \Delta F, \Delta N) = 2\pi\sqrt{\frac{\lambda_{hist}^2 + 2}{\det(A)}} \exp\left\{\frac{1}{2}\left(J^T A^{-1} J - C\right)\right\}$$

where

$$A = \begin{bmatrix} \sigma_T^{-2} + \lambda^2\sigma_N^{-2} & \lambda\sigma_N^{-2} \\ \lambda\sigma_N^{-2} & \sigma_F^{-2} + \sigma_N^{-2} \end{bmatrix}$$

$$J = \begin{bmatrix} \mu_T^2\sigma_T^{-2} + \lambda_{hist}\mu_N\sigma_N^{-2} \\ \mu_F^2\sigma_F^{-2} + \mu_N\sigma_N^{-2} \end{bmatrix}$$

and

$$C = \mu_T^2\sigma_T^{-2} + \mu_F^2\sigma_F^{-2} + \mu_N^2\sigma_N^{-2}$$

## A2   Likelihood vs Probability

We note that this is method is distinct from estimating $\lambda$ as the ratio of the distributions $\Delta F$ and $\Delta T$. This is due to a conceptual difference between probability and likelihood. Constructing the likelihood answers the question, "(a): how likely is a particular hypothesis (in this simple case, a particular value of $\lambda$) given the evidence?" This is a fundamentally different question from "(b): what is the probability density function of the ratio $-\Delta F/\Delta T$?" The first question involves fixing a putative value of $\lambda$,





which is *not* treated as a random variable. The second question treats $\lambda$ as a random variable. Mathematically, this is reflected in the difference between a line integral over the curve $y = -\lambda x$:

$$(a) : P(x,y|\lambda) = \int_C P_{xy}(x,y)ds = \int_{-\infty}^{\infty} P_{xy}(x,-\lambda x)\sqrt{1+\lambda^2}\,dx$$

and the ratio distribution of the random variable $\lambda = -y/x$

$$(b) : P_\lambda(\lambda) = \int_{-\infty}^{\infty} P_{xy}(x,-\lambda x)|x|\,dx$$

We use the ratio distribution (b) to estimate S once we have the posterior PDF for $\lambda$. This is because we treat S as the ratio of two random variables $F_{2xCO_2}$ and $\lambda$.

## A3   Correlations between $F_{2\times CO_2}$ and $\Delta F$

$CO_2$ emissions are the primary contributor to present-day radiative forcing change relative to preindustrial. Atmospheric concentrations of $CO_2$ were lower in the Last Glacial Maximum. This means that the forcing terms $\Delta F$ used as evidence in the LGM and historical periods are correlated with the forcing corresponding to doubled $CO_2$. For visual clarity, we neglect this correlation in this paper. To take it into account, we can write the simple energy balance model as

$$\Delta N = \Delta F' + \beta F_{2\times CO_2} + \lambda \Delta T.$$

In this case, the likelihood $P(E|\lambda, F_{2\times CO_2})$ is defined as the integral of the joint probability distribution of the evidence $E$ over the curve defined by the model. Following S20, we can then calculate $S$ by changing variables and marginalizing over $F_{2\times CO_2}$

$$P(S|E) = \int P(\lambda', F'_{2\times CO_2}|E)\delta(S - F'_{2\times CO_2}/\lambda')(\partial S/\partial \lambda')^{-1}(\partial S/\partial F'_{2\times CO_2})^{-1}dF'_{2\times CO_2}d\lambda'$$

Practically, we can draw samples of $\lambda$ and $F'_{2\times CO_2}$ from the joint posterior distribution and use these to calculate a posterior distribution for $S$. This correlation contributes very little to the results; when taking it into account we obtain similar ranges for $S$ as when we neglect it.

*Author contributions.*   KM and MW conceptualized the analysis. KM wrote the paper, with extensive input from MW. KM performed the analysis and wrote software, with assistance from MW.

*Competing interests.*   The authors declare that no competing interests are present.



*Acknowledgements.* KM wishes to acknowledge helpful discussions with Gavin Schmidt.



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
