# Peer review of "Towards robust community assessments of the Earth's climate sensitivity"

_EGUsphere, 2024_

## Referee Comment (RC1)

**Review of *Towards robust community assessments of the Earth's climate sensitivity**

**General Comments**

The authors present a nice overview of how Bayesian statistics can be used to make an assessment on climate sensitivity. Importantly, they discuss the various choices taken to make inference on climate sensitivity and how the choices affect the resulting estimate. While the paper is written well and most information is presented well, I feel the key concepts are somewhat obscured by their notation and lack of clarity. Below, I include some specific examples that I feel could be improved upon.

**Specific Comments**

1. Numerous terms are undefined. For example, in Eq. (1), what is $\mathcal{M}_0$? (It is not defined until two pages later). Same with $\mathcal{M}_\alpha$ and other models. Also, while it may be colloquial for some, $F_{2\times CO2}$ and $\Delta T_{2\times CO2}$ are undefined.

2. Various mathematical terms are undefined.

   - Line 105 - What is $N(-8.43, 2)$? I assume you mean a normal distribution with mean $-8.43$ and variance 2. Also, math cal font is used for N in section 7 and not for line 105.
   - Line 201 - What is $\Delta T'(x)$ and what is $x$?

3. In multiple areas the authors discuss the idea of reducing uncertainties but do not explicitly say which uncertainties are being (or will be) reduced. I think this is a common misnomer when discussing statistical concepts. Some uncertainties are irreducible, such as types of data uncertainty, and some are reducible, such as types of model uncertainty. A key concept in statistics is being able to identify each type and providing the best possible quantification of each - e.g. an appropriate quantification of irreducible uncertainty and reducing all other uncertainty if possible. I feel strongly that the manuscript would benefit if this distinction were made clear.

4. Line 60 - "... update our prior beliefs $P(\Theta)$... " Are you making the distinction that you can update you $P(\Theta|Y)$ if new information becomes available? If so, I think this needs to be reworded. Otherwise, I do not believe Bayes' Theorem says update our prior beliefs. Instead, if you have a prior belief or knowledge, $P(\Theta)$, you can get an estimate of the probability of $\Theta$ given data/evidence $Y$ using that prior knowledge. If more data becomes available, you could then refine that belief and use a new prior. This is distinct from updating your prior belief.

5. Section 2 Analysis framework - This is a crucial section for your paper and I would like to see it expanded. Throughout the rest of the paper, terms like *posterior*, *marginal*, *joint probability density*, ... are used but not defined. The general reader of ESD may be unfamiliar with these terms. The latter parts of the paper would be easier to follow if these terms are defined in the context of section 2. Additionally, how does one get $P(\Theta|Y)$? Is it as simple as writing distributions down and using Bayes' Theorem? What if the distribution is not tractable, how would that be handled? Expanding on some of the steps needed to make inference on $\Theta$ within

this section will help orient readers as to why this is such a difficult and important problem, and how they can take what you have shown and apply it to their own analysis.

6. Line 120 - The notation surrounding this equation is confusing. It appears as though you are treating $Y = (\Delta T, \Delta F)$ as normally distributed random variables where the mean and standard deviation of each are estimated from experts. You define the joint probability density of $\Delta T$ and $\Delta F$ as $\mathcal{J}(Y) \equiv \mathcal{J}(\Delta T, \Delta F)$. You then *marginalize* over $\Delta T$ and $\Delta F$ and somehow get a likelihood of the evidence as $P(\Delta T, \Delta F | \lambda)$. However, this equation (A.3), does not contain either $\Delta T$ or $\Delta F$ because they have been marginalized out. Instead it contains their mean and standard deviation (assumed fixed?) that are estimated by experts. My confusion is in your definition of evidence and how $\Delta T$ and $\Delta F$ (or their mean and standard deviation) play a role in that evidence. I think this could be fixed by being more clear on your notation and the steps taken to arrive at the equation on line 120 (and subsequent equations).

7. Model $\mathcal{M}_\alpha$ - This is slightly confusing to me. By definition, $\alpha = \partial\lambda/\partial\Delta T$ is a function of $\lambda$. However, you assign independent priors to $\lambda$ and $\alpha$ when $\alpha$ is constrained by the value of $\lambda$. Is there justification for specifying independent priors here? Or is this done for illustrative purposes? If so, I feel it is important to note they are not independent.

8. Eqn. (7) - Same as above comment except now for $\Delta\lambda$.

9. Section 7.1 - I rather like this section and I think it puts a lot of the paper into context. However, I feel as though some terms are not defined and potentially unknown to the general ESD reader. A (Bayesian) hierarchical model is left undefined and for the reader to interpret. Generally, a BHM is defined in terms of data, parameter, and sometimes hyperparameter models. It might help contextualize your message if you state what a hierarchical model is in terms defined from section 2 and then connect it to equations (9)-(12).

10. Line 261 - This is a very bold claim. I would argue it is *one* useful application of hierarchical modeling, but maybe not one of the most.

11. Line 396 - $C$ is already defined as the curve.

**Technical Comments**

1. A few citations have typos in or surrounding how they are placed in the text

2. Figure 1 - needs labels

3. Line 9 - multiple twice

4. Line 234 - Missing )

5. Line 241 - "... that this is are the ..."

6. Line 243 - Missing )

---

## Community Comment (CC1)

11 March 2024

**Review of**
**"Towards robust community assessments of the Earth's climate sensitivity"**
**by Kate Marvell and Mark Webb.**
**Preprint. Discussion started: 22 February 2024**

**General Comments**

1.      This is an interesting paper, containing both results and discussion of method that are likely to be helpful to the community involved in assessments of climate sensitivity.

2.      I suggest that the paper could be improved substantially by adopting different terminology: by transferring from a terminology appropriate to a subjective interpretation of probability theory (flowing from a subjectivist theory of knowledge) to a terminology appropriate to objective theories of both probability and knowledge. This would involve no changes to the equations or the results, as the mathematics of the probability theory would be unchanged, but it would change the way in which the mathematics is interpreted in terms of its relation to the real world.

3.      Specifically, I suggest the term "belief" (particularly in the term "prior belief") be changed throughout. In most place it could be replaced by "estimate" or "information" or "knowledge". In other places the meaning is different, and it would be better replaced by "assumption". Similarly, I suggest that the term "subjective" is over-used. In most places, what is described as "subjective" is in fact objective, i.e. it is inter-subjectively shared and criticised. In most cases, this sharing and criticism is of the very high standard expected of publications in the scientific literature.

4.      So, if accepted, these comments would imply numerous changes to the text, but ones that could be made without changes to the structure and scientific content of the paper.

5.      A subjective theory of knowledge was widely accepted up to the middle of the 20th century. It was accompanied by a subjective interpretation of probability theory in general and of Bayes theorem in particularly. This interpretation was heavily criticised by Karl Popper in many of his key works. The preference for an objective rather than a subjective, theory of probability is discussed most cogently by Popper in "Realism and the aim of science" (1983). Chapter 1 of Part II is entitled "Objective and subjective probability", and the comments in this review are intended to be consistent with Popper's treatment of these problems. In summarising the difference between these two approaches to probability theory, Popper says (section 7, para 1): "… The subjectivist takes $a$ as his hypothesis and $P(a|b)$ as our degree of belief in it, whilst the objectivist takes '$P(a|b)=r$' as his hypothesis. (He may or may not believe in it.) …" . (The subjectivist example here stands for the probability that hypothesis $a$ is true given evidence $b$, but it applies equally to the case where $a$ is the estimate of a quantity and $b$ is the observational evidence supporting it.)

If one accepts Popper's criticisms, then the subjective interpretation of probability is both out-moded and unnecessary (although it appears to linger on in some text books on philosophy of science and on statistics).

6.      One could argue that we should not worry about words, because "belief" could be interpreted as "estimate" or "information" or "knowledge" or "assumption". However, I suggest that it is unhelpful to use "belief" in a way that differs radically from its everyday usage. This is epitomised by the biblical story of Doubting Thomas: "Jesus saith unto him, Thomas, because thou hast seen me, thou hast believed: blessed *are* they that have not seen, and *yet* have believed." (John, 20: 29). I suggest that in science, we tend to side with

Thomas rather than with Jesus - we tend to demand the evidence and to avoid belief without it.

7. Another problem of using the term "prior belief" for an element on the right-hand side of the Bayesian equation is that, if we are consistent, the term on the left-hand side of the equation is then a "posterior belief". However, in this paper and elsewhere, the implication is that the result of the Bayesian process is an objective result, rather than just a belief – that, somewhere along the line, a subjective belief is transformed into an objective estimate. Objective theories of probability avoid this problem.

8. There is much reference in the paper to "expert judgement" but expert judgement is informed by past experience and its accompanying evidence. Moreover, it is not derived subjectively but through participation in the objective work of the scientific community.

9. I think the only example of "belief" in this paper is where an "expert" persists in making a judgement despite evidence to the contrary. I think this is rare – usually there is objective evidence for a judgement, even though the evidence is incomplete. A good scientist recognises that it is incomplete and is open to new evidence.

10. More generally, a good scientist holds his/her views tentatively and hypothetically, recalling that scientific progress takes place through the replacement of one false hypothesis by a better (but probably false) hypothesis. Consequently, a good scientist tries not to "believe" anything but to work via a series of hypotheses and assumptions and their testing.

**Detailed Comments**

11. l.7. Here and many other places. "beliefs". See General Comments above.

12. l.9. Here and many other places. "subjective". See General Comments above.

13. l.19, eq.(1). What is $M_0$? - the climate system, a model of the climate system, or the simple energy balance model? If the last, then is the RHS of (1), i.e. including $\Delta N$, different?

14. l.33: "aerosols". Net cooling in response to aerosols?

15. l.49 and l.170: "knowledge". See General Comments above.

16. l.61-622, eq.(3) and following line. If $P(\Theta)$ is a belief, then $P(Y|\Theta)$ must also be a belief (a posterior belief). See General Comment 7 above.

17. Fig.1(a). Axes need labels.

18. l..120, equation. This is not very clear. C is not defined.

19. l.161-162. Sentence "These incorporate expert judgement …". These are normally objective, not subjective, i.e. they are inter-subjectively shared and criticised. This is fundamental in science.

20. l.166-167: "well-informed scientist". Again, informed by objective information.

21.    I.217-218: "Why do these two distributions not overlap substantially?" They appear to overlap substantially - they are well within each other's one-sigma points.

22.    I.229: "odds". This is another word associated with a subjective theory of probability, and best avoided if you adopt an objective approach.

23.    I.237: "definite". What does definite mean here? Does it mean "certain"? If so, this would not be a scientific statement - uncertainty is all-pervasive in science. If you remove "definite" from this sentence, do you not conclude that state dependence is likely?

24.    I.242: "We are *not* arguing that this is the objectively "correct" way to combine the Last Glacial Maximum reconstructions with historical observations." Given my comment above, it is not clear what you are arguing here.

25.    I.250: "relying on a community of experts". Yes! - this makes it objective - this is how we do science - inter-subjectively shareable and criticisable.

26.    I.263: "assume". Yes - so these are hypotheses (to be tested), not beliefs.

27.    I.286: "prior assumptions: Yes - much better! You can assume something without believing it.

28.    I.290: "accurate". Meaning exact? Unusually, accuracy means a quantification of the uncertainty.

29.    I.294: "belief". At no point in the discussion contained in this paragraph do you need to "believe" anything - you are making certain assumptions or posing certain hypotheses, and then testing their consequences.

30.    I.337: "the prior beliefs of two hypothetical experts". Or you could say just two hypotheses?

31.    I.341: "However, some experts may not be so open-minded …". So, are you are saying that there are closed-minded experts who "believe" things and open-minded experts who make hypotheses?

32.    I.343-345: "Expert C's confidence remains unshaken …" and following sentence. This is fundamental to how science works. You are saying that Expert C is not influenced by evidence and so is not behaving rationally/scientifically. In (good) science, we suspend belief and act tentatively and hypothetically.

33.    I.348-349: "The narrowness of C's prior …". It's OK to have a narrow prior, if all the evidence you have (at present) points in that direction, but it is prudent to assume that there is some possibility (low probability) of a gross error, because of some effect that has not been handled correctly. This leads to a different pdf (e.g. Gaussian + constant). This is a common problem is science - it's called quality control.

34.    I.379: "The beauty of Bayesian methods …". The beauty of objective Bayesian methods is that you don't need to deal in "belief" at all.

35.     l.396, eq.(A3).  Superficially, there appears to be a minus sign missing here – required for a Gaussian shape.

36.     Equations at the end of section A1.  Again, not clear how this leads to a Gaussian shape.

**Editorial Comments**

37.     l.242: typo "is are".

38.     l.260, 261.: typos "s(S", "y(K"

39.     A2, line 1: typo "this is method is distinct".

---

## Author Response (AR1)

We thank the reviewers for their thoughtful comments, which have very much improved our paper!  Please find attached a "tracked changes" version of the manuscript.

R1

Review of Towards robust community assessments of the Earth's climate sensitivity

General Comments
The authors present a nice overview of how Bayesian statistics can be used to make an assessment on climate sensitivity. Importantly, they discuss the various choices taken to make inference on climate sensitivity and how the choices affect the resulting estimate. While the paper is written well and most information is presented well, I feel the key concepts are somewhat obscured by their notation and lack of clarity. Below, I include some specific examples that I feel could be improved upon.

Specific Comments

Numerous terms are undefined. For example, in Eq. (1), what is $M_0$? (It is not defined until two pages later). Same with $M_\alpha$ and other models. Also, while it may be colloquial for some, $F_{2\times CO2}$ and $\Delta T_{2\times CO2}$ are undefined.

We agree!  We've now defined these terms.
Various mathematical terms are undefined.

• Line 105 - What is $\mathcal{N}(-8.43, 2)$? I assume you mean a normal distribution with mean $-8.43$ and variance 2. Also, math cal font is used for $N$ in section 7 and not for line 105.

Fixed.  As an aside, thank you for calling this out.  Often in the literature it's unclear whether s is the standard deviation or the variance in the notation N(m,s).

• Line 201 - What is $\Delta T'(x)$ and what is $x$?

This has been removed.

In multiple areas the authors discuss the idea of reducing uncertainties but do not explicitly say which uncertainties are being (or will be) reduced. I think this is a common misnomer when discussing statistical concepts. Some uncertainties are irreducible, such as types of data uncertainty, and some are reducible, such as types of model uncertainty. A key concept in statistics is being able to identify each type and providing the best possible quantification of each - e.g. an appropriate quan- tification of irreducible uncertainty and reducing all other uncertainty if possible. I feel strongly that the manuscript would benefit if this distinction were made clear.

We've now tried to make this much clearer.  This paper is intended as a guide for those embarking on large-scale expert assessments, whether a follow-up to Sherwood et al 2020 or assessments of other quantities like TCR or ZEC.  To that end, we're focusing on identifying the areas where unavoidable subjective decisions (i.e., expert judgment) enter the analysis.

Line 60 - "... update our prior beliefs P(Θ)... " Are you making the distinction that you can update you P(Θ|Y) if new information becomes available? If so, I think this needs to be reworded. Otherwise, I do not believe Bayes' Theorem says update our prior beliefs. Instead, if you have a prior belief or knowledge, P(Θ), you can get an estimate of the probability of Θ given data/evidence Y using that prior knowledge. If more data becomes available, you could then refine that belief and use a new prior. This is distinct from updating your prior belief.

Yes, this is a tough one to convey.  We were trying to balance technical rigor with comprehensibility for a wider audience.  Please see the rewritten "Analysis Framework" section:

"This framework allows us to use our prior understanding of the parameter values to calculate the posterior probabilities P (Θ|Y, M ) of the model parameters given the evidence. This posterior can be updated as new evidence becomes available."

Section 2 Analysis framework - This is a crucial section for your paper and I would like to see it expanded. Throughout the rest of the paper, terms like posterior, marginal, joint probability density, ... are used but not defined. The general reader of ESD may be unfamiliar with these terms. The latter parts of the paper would be easier to follow if these terms are defined in the context of section 2. Additionally, how does one get P (Θ|Y )? Is it as simple as writing distributions down and using Bayes' Theorem? What if the distribution is not tractable, how would that be handled? Expanding on some of the steps needed to make inference on Θ within
this section will help orient readers as to why this is such a difficult and important problem, and how they can take what you have shown and apply it to their own analysis.

We've substantially rewritten the entire section for clarity- see the revised text below. The section now begins with a definition of evidence, model, and prior.  (Note that now we explicitly represent the prior dependence on the model).  We then define prior, likelihood, and posterior as relevant for the problem of assessing ECS.

"Bayes' Theorem can be written as

$$P (\Theta|Y, M ) = P (Y |\Theta, M ))P (\Theta|M )/P (Y |M ) . \quad (3)$$

Here, we will define these terms as they apply to the problem of estimating climate sensitivity.

Evidence The evidence Y used to constrain climate sensitivity consists of the global mean

temperature change $\Delta T$ in response to a forcing $\Delta F$ as well as, in non-equilibrium

states, the net energy imbalance $\Delta N$ . We have estimates of these quantities for the

historical period (derived from observations and models) and for past climate states

(derived from paleoclimate proxies and models), and Y therefore consists of multiple lines $Y_1 \ldots Y_n$. For example, S20 used process-based understanding of underlying

physics, recent observations, and proxy-based reconstructions of past climates to assess

S.

The model M codifies how we interpret the evidence Y . It specifies the parameters

$\Theta$ whose posterior distributions we estimate. For example, in the simple energy balance

model denoted M0, there is only one parameter and $\Theta = \lambda$. The model determines the

likelihood $P (Y |\Theta, M )$ of observing the data given particular values of the parameters

$\Theta$. We discuss methods for calculating this likelihood in Section 4.1.

The prior probability distribution $P (\Theta|M )$ reflects prior beliefs or knowledge about

the model parameters $\Theta$. For example, in the simple model M0, the community assessment S20 adopted a uniform prior on $\lambda$ as a baseline choice, choosing not to rule out

net positive feedbacks (and therefore an unstable climate) a priori. "

Line 120 - The notation surrounding this equation is confusing. It appears as though you are treating Y = ($\Delta T,\Delta F$) as normally distributed random variables where the mean and standard deviation of each are estimated from experts. You define the joint probability density of $\Delta T$ and $\Delta F$ as J(Y) $\equiv$ J($\Delta T,\Delta F$). You then marginalize over $\Delta T$ and $\Delta F$ and somehow get a likelihood of the evidence as P($\Delta T,\Delta F|\lambda$). However, this equation (A.3), does not contain either $\Delta T$ or $\Delta F$ because they have been marginalized out. Instead it contains their mean and standard deviation (assumed fixed?) that are estimated by experts. My confusion is in your definition of evidence and how $\Delta T$ and $\Delta F$ (or their

mean and standard deviation) play a role in that evidence. I think this could be fixed by being more clear on your notation and the steps taken to arrive at the equation on line 120 (and subsequent equations).

You're right that this section was lacking clarity. In climate sensitivity assessments, the evidence is necessarily uncertain. We don't have point measurements of temperature, forcing, etc. Instead, we can assess the literature and come up with a joint probability density $\rho(\Delta T, \Delta F)$ (here, this is the product of the PDFs for $\Delta T$ and $\Delta F$, but more complex distributions reflecting correlated errors are possible). In this paper, we make the argument that given a model M described by a curve C in evidence space, the likelihood can be approximated by calculating the probability mass along the curve. S20 calculated historical and paleo likelihoods using a similar method but different language; to our knowledge no other climate statistics paper has employed the probability mass method. We've rewritten the section to be clearer.

Model $M\alpha$ - This is slightly confusing to me. By definition, $\alpha = \partial\lambda/\partial\Delta T$ is a function of $\lambda$. However, you assign independent priors to $\lambda$ and $\alpha$ when $\alpha$ is constrained by the value of $\lambda$. Is there justification for specifying independent priors here? Or is this done for illustrative purposes? If so, I feel it is important to note they are not independent.

In this case, $\alpha$ is a constant, not a continuous function of $\lambda$. There is no a priori reason why the uncertainty in the net feedbacks should be correlated with the uncertainty in their rate of change with global mean temperature. The point here is to make N a quadratic function of $\Delta T$, $N = \Delta F + a\Delta T + b\Delta T^2$ where $a = \lambda$ and $b = \alpha/2$.

Eqn. (7) - Same as above comment except now for $\Delta\lambda$.

$\Delta\lambda$ is a Gaussian that we specify, which is independent of $\lambda$ _hist. Obviously the fact that the net feedback $\lambda = \lambda$ _hist + $\Delta\lambda$ means that $\lambda$ is not independent of $\lambda$ _hist or $\Delta\lambda$? Perhaps this is where the confusion arises?

Section 7.1 - I rather like this section and I think it puts a lot of the paper into context. However, I feel as though some terms are not defined and potentially un- known to the general ESD reader. A (Bayesian) hierarchical model is left undefined and for the reader to interpret. Generally, a BHM is defined in terms of data, pa- rameter, and sometimes hyperparameter models. It might help contextualize your message if you state what a hierarchical model is in terms defined from section 2 and then connect it to equations (9)-(12).

We've rewritten the section for improved clarity, and no longer provide a meta-analysis estimate of radiative forcing, which we now realize enlarged the scope of our paper unnecessarily. To appeal to a broader audience, we've also de-emphasized the

hierarchical nature of Bayesian meta-analysis, stressing the physical meaning of the priors on the hyperparameters mu and tau.  This has also helped us make clearer recommendations: these priors must be specified, and we suggest that organizers of future assessments do so (as opposed to querying the broader community of experts).

Line 261 - This is a very bold claim. I would argue it is one useful application of hierarchical modeling, but maybe not one of the most.

Removed.

Line 396 - C is already defined as the curve.

fixed

Technical Comments

1. A few citations have typos in or surrounding how they are placed in the text 2. Figure 1 - needs labels

fixed.
 3. Line 9 - multiple twice fixed
 4. Line 234 - Missing ) This discussion has been removed from the paper.

5. Line 241 - "... that this is are the ..." 6. Line 243 - Missing ) This discussion has been removed from the paper.

R2

This manuscript presents an assessment of the uncertainty associated with Bayesian inferences of the effective climate sensitivity parameter, S as assessed by Sherwood et al. (2020).  The authors clearly point out three sources of uncertainty that were not previously addressed in Sherwood et al. (2020): evidence uncertainty, structural uncertainty, and prior uncertainty and illustrate each with examples related to the Last Glacial Maximum.  They conclude with a recommendation of how to combine multiple lines of evidence to constrain S that will allow for rapid updates in light of new evidence in the future.

The manuscript creates more awareness in terms of the uncertainty involved in Bayesian inferences such as that of Sherwood et al. (2020).  The topic is both

interesting and important. The authors provided several interesting examples, however, in my opinion, the manuscript reads somewhat esoteric for the atmospheric science community. Overall, I have a few suggestions that are quite minor in nature, mostly regarding clarifying the main messages for individual sections for the authors to consider before recommending publication.

For context, it would be helpful if the authors could please specify in the Introduction the uncertainty that was discussed in Sherwood et al. (2020) and then follow this with the additional complementary detail that they consider.

This is a helpful comment. We've now tried to clarify the aims of our paper vs those of S20 (and IPCC AR6) and have added the text below.

 IPCC AR6 assessed confidence in the range of S based on suport from individual lines of evidence, and the medium confidence assessed was in large part due to the fact that not all lines of evidence supported the same upper bound. By contrast, S20 sought to provide a robust estimate by combining lines of evidence in a coherent Bayesian framework. However, S20 used baseline priors and estimates of the evidence and investigated the impact of alternate choices as sensitivity tests rather than attempt to combine multiple priors, estimates, and expert judgements into a single posterior probability distribution. In both IPCC AR6 and S20, as in almost all previous assessments, the means by which disagreements among experts were resolved or handled was not necessarily made transparent. This paper presents some lessons learned by two authors of S20 and attempts to chart a way forward.

Our goal is not to provide a single updated estimate but rather to understand where unavoidable subjective decisions enter in to the analysis and to present a framework for systematically and fairly incorporating the subjective judgements of multiple experts.

Section 6.1: this section on comparing "apples to apples" when considering different lines of evidence to constrain S is an important point that was raised, and the example is interesting, however, the main point is unclear and the description is too roundabout. In this particular example, are the authors recommending the Bayes factor as a solution for evaluating the reliability of different lines of evidence? I would suggest a clear statement at the end of this section regarding the authors' recommendation regarding how to treat the issues of fairly comparing different lines of evidence when constraining S.

This section was initially quite confusing, and we apologize. We've rewritten it to stress the importance of the interpretive model in rendering multiple lines of evidence compatible and resolving the "Twin Peaks" problem. The "best" model will depend on prior knowledge: if we have reason to believe that one interpretive model is better than another, then the Twin Peaks problem may not be an actual problem: a small area of

overlap between posteriors updated with multiple lines of evidence may then constrain the parameters extremely well. However, if we have no reason to believe that one model is "better" than any other, then models that render lines of evidence more compatible will be preferred.

Lines 248-251: This brings up the important point that subjectivity is an issue, however, other than suggesting more transparency in terms of making subjective decisions, it does not seem that the authors are reducing any subjectivity. Please clarify. For example, on lines 319-320, why use a weighted average? This choice itself is apparently subjective. A well-justified recommendation could better convince others to follow these recommendations in the future.

We've clarified to explain that we're not necessarily reducing subjectivity: as Figure 1 shows, there are unavoidable subjective decisions in every analysis. Instead, we're arguing that these decisions need to be clearly communicated, and that expert judgment should be clearly specified in the form of priors. We've now added specific "recommendation" sections throughout Section 7 to make this clearer. In short, we present a method for achieving transparency and clarity on necessary subjective decisions made.

I recommend that the authors also summarize their specific recommendations for future Bayesian analysis in the Abstract. There is space for it in the Abstract.

Good suggestion, done.

Typographic errors

> Lines 242: "that this is are the model…" should be "that this is the model…"
> This discussion has been removed from the paper.
> Line 269: "about a some true…" should be "about some true…" fixed
> Line 279: "As an specific" should be "As a specific" fixed
> Line 329: Latex compilation error for Table fixed, table moved to supplementary material
> Line 343: "distributions" should be singular fixed
> Line 337: Isn't Expert B also open to the idea that lambda can be positive too in Figure 8a?
> For consistency and convenience, please number all equations, including the ones on page 13, even if not explicitly referenced in the text. fixed
> The notation N(x,y) was defined in Sherwood et al. (2020) but not in this manuscript. Please define it in this manuscript. defined.

R3

In this manuscript the authors discuss issues and possible ways forward with bayesian based assessments of climate sensitivity. This follows on a first major attempt by Sherwood et al. (2020), which was influential on the IPCC report (Forster et al. 2021). I have no major issues with the manuscript, and I think it is great that the approach and issues are discussed openly. I would have hoped, before reading it, that the text would have been even more accessible to a wider audience, but in several places there is quite a bit of statistics jargon.The minor recommendations below are not fully addressing this issue, and I would leave it up to the authors to consider this issue. Anyway, I see no major obstacles to publication.
* * *
2-3, Most of these lines of evidence constrain S or ECS, not feedback in isolation.

We've replace this with "The uncertainty inS primarily results from uncertainties in net physical climate feedback, usually denoted as \lambda. "

26-27, Note that IPCC did not use the bayesian method, so perhaps state what they did. Furthermore, the medium confidence is due to not all individual lines of evidence supporting a 95th percentile close to 5 K.

We've now noted this in the Introduction- but note that assessing confidence based on support from individual lines of evidence is at odds with our proposed method of combining lines of evidence in a coherent Bayesian framework.

26, the reference should be Forster et al. (2021) fixed

53, remove one instance of 'is'

Removed

59, why use quotation marks here?

Removed

63-64, I am not sure this was what the lines of evidence were called in S20.

It wasn't, but we've adopted this notation for clarity.  Note the rewritten Analysis section that now clearly defines all terms.

96-97, See also Annan et al. (2022) for a discussion of how T20 might be cold-biased and over-confident due to reliance on a single-model prior.

Agreed.  We've now noted this and emphasized that the two studies are not comparable, and use them only to illustrate the impact of evidence uncertainty

98, These are not simply two equally valid or comparable studies. S20 is an assessment in which, in principle, the authors took into account a much broader evidence base than used by T20.

Agreed- see response above.

105, should probably be '-9.6'

Fixed

125, missing closing parenthesis.

Fixed

Section 4, Perhaps check out https://doi.org/10.5194/cp-19-323-2023.

Cited.

136-137, The quadratic model is that ECS changes monotonically until an instability occurs. There is some evidence that ECS will increase with warming, but we also know there were snow ball Earth instabilities in the past, so ECS must increase into a bifurcation at cooling temperatures. The shape of this function is not well known, other than that there is a minimum not too far from our current climate. What I am getting at is that the quadratic model is only half the story, and the evidence for a positive alpha comes mostly from warmer climates. It might be negative at colder temperatures, which effectively means the model is no longer valid.

This is of course possible, and we've now alluded to the potential bifurcation of alpha with temperature in our list of potential models.

166-167, A bit of an understatement. Since we are here it is not physically possible that climate sensitivity is negative, i.e. the system is unstable, so prior knowledge forbids negative climate sensitivity.

In S20, flat uniform priors were placed on the individual feedback components in the process evidence, implying a prior on the net feedback that puts equal weight on positive and negative values. Clearly, as you point out, this is unphysical. S20 dealt with this by removing low-likelihood values of individual feedback components, while here we prefer to allow unstable climates to be definitively ruled out by the evidence. In the prior uncertainty section we do use the knowledge derived from the process evidence as an alternate prior.

Figure 3, I think this would have been useful earlier, just a thought.

Thanks! We've now moved it to Figure 1.

198, + Modak and Mauritsen https://doi.org/10.5194/acp-23-7535-2023

Cited.

203, Note that this is based on the AMIPII dataset which produces the largest pattern effect of all SST reconstructions (https://doi.org/10.5194/acp-23-7535-2023)

Yes, we've now noted that the prior on Delta lambda used in S20 may be too weighted toward large values (and possibly too narrow), but we use the S20 marginal historical likelihood for illustrative purposes.

223-224, I find this argument weak: it looks better, hence it must be right? On could also state that we trust historical warming, and we will use alpha as a fudge factor on the LGM evidence to make it match the historical.

We've substantially rewritten this section, but we do feel that the "Twin Peaks" problem is worth noting.  If posteriors updated by multiple lines of evidence have a small region of overlap, one of two things is true: either we can be highly confident in the resulting estimates, or the lines of evidence are not, in fact, measuring the same thing.  In the absence of prior knowledge regarding either of these possibilities, a model that brings posteriors from different lines into better agreement has more evidence to support it.  If we do have prior knowledge or beliefs that we are indeed comparing "apples to apples", this is reflected in the term P(M1)/P(M2), and causes the resulting posterior derived from multiple lines of evidence to be more sharply peaked.

Equation above 235 and the equation shortly thereafter, there is a missing closing parenthesis.

This has been removed.

244 (some issue with line numbering, that is the line just above 245), A strange formulation, I would say "If using T20 evidence, more agreement with historical evidence is obtained if assuming alpha is close to zero." If one were to use Annan et al. (2022) or a weaker pattern effect estimate, then the result would be different.

We agree completely with this statement.  We've now rewritten the text to say "Clearly, the ``best" model depends on the evidence used, the prior knowledge of whether we are comparing ``apples to apples", and the priors we place on $\lambda$, $\Delta \lambda$, and $\alpha$."

288, Perhaps comment why this entire range is warmer than the range estimated above? It is the same evidence, I suppose, so a nice example of how a too wide asymmetric prior can bias the posterior.

We've added "and warmer"- note that this is because a "fixed effects" model will simply treat cool estimates as outliers, hence the warmer values.

339, missing table number

Table deleted in rewrite.

Table 3, '1,9' -> '1.9'

Table deleted in rewrite.

342, The authors are careful to write she/he in other places, but not here. Why is it that the extremely overconfident expert is male?

We've added she pronouns.

Section 8, I felt this section was hard to read, and I feel like it could be shortened and sharper.

We've made some big changes, not least identifying specific recommendations.

353, "where do those estimates or measurements come from", I think this text is open to too much interpretation.

We've replaced this with "how do we decide what counts as "evidence", which we hope is clearer.

356, please state which section.

This has been rewritten to describe the overall strategy.

371, delete one instance of 'make'
Done.

Comment by John Eyre

**General Comments**

1. This is an interesting paper, containing both results and discussion of method that are likely to be helpful to the community involved in assessments of climate sensitivity.

Thank you.

2. I suggest that the paper could be improved substantially by adopting different terminology: by transferring from a terminology appropriate to a subjective interpretation of probability theory (flowing from a subjectivist theory of knowledge) to a terminology appropriate to objective theories of both probability and knowledge.  This would involve no changes to the equations or the results, as the mathematics of the probability theory would be unchanged, but it would change the way in which the mathematics is interpreted in terms of its relation to the real world.

In Bayesian statistics there are multiple schools of thought, including subjectivist Bayesianism and Objectivist Bayesianism (see Gelman and Hennig 2017 for a review). There is a lack of consensus on the best terminology to use in Bayesian statistics, with many different approaches being advocated by different researchers. Additionally, Gelman and Hennig (2017) argue that the words 'objective' and 'subjective' in statistics discourse are used in a mostly unhelpful way.  Many of your points certainly might be valid, but we are not experts on the philosophy of statistics, and the purpose in this study is not to address longstanding debates about the terminology of Bayesian statistics.  It is to propose improvements to the way that Bayesian statistics can be applied to the problem of organizing community assessments of evidence.

Our understanding is that subjectivist language is more commonly used than objectivist language in the literature and; the terminology we have adopted is commonly used if not universally agreed upon.  Perhaps more importantly, the colloquial understanding of Bayesian methods (such as it exists) regards Bayesian inference as analogous to a learning process, in which prior beliefs are replaced by updated beliefs in light of evidence.   That said, we find your arguments compelling and  will also adopt some of your specific proposals - please see below for details.

3. Specifically, I suggest the term "belief" (particularly in the term "prior belief") be changed throughout.  In most place it could be replaced by "estimate" or "information" or "knowledge".  In other places the meaning is different, and it would be better replaced by "assumption".  Similarly, I suggest that the term "subjective" is over-used.  In most places, what is described as "subjective" is in fact objective, i.e. it is inter-subjectively shared and criticised.  In most cases, this sharing and criticism is of the very high standard expected of publications in the scientific literature.

We've replaced most instances of "belief" with "belief and/or knowledge" or just "prior". However, in some cases, we humbly suggest that we do mean "belief" in the actual sense.  See, e.g., our discussion of how to handle experts with overly narrow and/or biased priors.  While scientific knowledge should, in theory, be updated systematically and dispassionately with evidence, it is not our experience in working with actual scientists that this always happens.  Scientists, as humans, approach questions with priors informed not just by previous evidence but emotional states, ego, cultural background, political biases, etc.  Our hypothetical scientist C strongly believes climate sensitivity to be low, not necessarily because s/he has extensive knowledge others with broader priors do not, but because s/he wants it to be.

4. So, if accepted, these comments would imply numerous changes to the text, but ones that could be made without changes to the structure and scientific content of the paper.

5. A subjective theory of knowledge was widely accepted up to the middle of the 20 th century. It was accompanied by a subjective interpretation of probability theory in general
and of Bayes theorem in particularly. This interpretation was heavily criticised by Karl

Popper in many of his key works. The preference for an objective rather than a subjective,
theory of probability is discussed most cogently by Popper in "Realism and the aim of science" (1983). Chapter 1 of Part II is entitled "Objective and subjective probability", and
the comments in this review are intended to be consistent with Popper's treatment of these
problems. In summarising the difference between these two approaches to probability theory, Popper says (section 7, para 1): "... The subjectivist takes a as his hypothesis and
$P(a|b)$ as our degree of belief in it, whilst the objectivist takes '$P(a|b)=r$' as his hypothesis.
(He may or may not believe in it.) ..." . (The subjectivist example here stands for the probability that hypothesis a is true given evidence b, but it applies equally to the case where
a is the estimate of a quantity and b is the observational evidence supporting it.)

If one accepts Popper's criticisms, then the subjective interpretation of probability is both out-moded and unnecessary (although it appears to linger on in some text books on philosophy of science and on statistics).

We're far from experts  in the philosophy of science and can't necessarily fault Popper here.  But more modern works (such as Gelman and Hennig 2017) discuss objectivist and subjectivist approaches and do not state that the subjective interpretation of probability is both out-moded and unnecessary.  Hence we must conclude that they do not wholly accept Popper's criticisms.  We recognize that this is an "appeal to authority" argument which may be flawed, but we don't feel sufficiently qualified to contribute to the objectivist vs subjectivist debate- we simply observethat the matter does not appear to be closed.

Popper (1984) is mainly concerned with to process of testing theories in physics. Again our use case is not limited to this - it's about estimating climate sensitivity.  Faced with a number of estimates of LGM cooling from different studies, how to we reach a consensus estimate when experts disagree on the merits of different studies? It's not clear exactly how to do this in an objectivist point of view, especially when other factors (rivalries, personalities, egos, biases) might enter in to it.  All we can do here is argue for transparency in decision making.

6. One could argue that we should not worry about words, because "belief" could be interpreted as "estimate" or "information" or "knowledge" or "assumption".  However, I suggest that it is unhelpful to use "belief" in a way that differs radically from its everyday usage.  This is epitomised by the biblical story of Doubting Thomas: "Jesus saith unto him, Thomas, because thou hast seen me, thou hast believed: blessed are they that

have not seen, and yet have believed." (John, 20: 29).  I suggest that in science, we tend to side with Thomas rather than with Jesus - we tend to demand the evidence and to avoid belief without it.

We are using it in sense of *rational* belief - see Belief, credence, and norms | Philosophical Studies (springer.com) This paper does not discuss religious belief.

We'd also refer to the Stanford Encyclipedia of Philosophy  entry on formal belief which discusses both subjective bayesian probability theory and personal questions of faith https://plato.stanford.edu/entries/formal-belief/  Perhaps see also https://plato.stanford.edu/entries/epistemology-bayesian/

7. Another problem of using the term "prior belief" for an element on the right-hand side of the Bayesian equation is that, if we are consistent, the term on the left-hand side of the
equation is then a "posterior belief". However, in this paper and elsewhere, the implication is
that the result of the Bayesian process is an objective result, rather than just a belief – that,
somewhere along the line, a subjective belief is transformed into an objective estimate. Objective theories of probability avoid this problem..

.We've removed instances of "posterior belief".  As another reviewer noted, we should not even regard the posterior as an "updated prior", simply the result of the Bayesian process given a set of priors and a model.

8. There is much reference in the paper to "expert judgement" but expert judgement is informed by past experience and its accompanying evidence.  Moreover, it is not derived subjectively but through participation in the objective work of the scientific community.

Expert judgment is of course informed by experience and evidence, and it would be ideal if experts  derived only via participation in the objective work of the scientific community.   As it stands, all experts could have more or less equal access to the published scientific literature- and yet disagreement would persist.  It is our goal here to propose methods for the world as it is, not necessarily as it should be.

Perhaps a useful way to think about this is in terms of a hierarchy of models.  Why don't we know LGM cooling?  We don't know which published estimate to believe.   We don't know the proper forward model that converts proxy reconstructions to global mean temperature.  There is uncertainty in the proxy measurements.  And so on and so on.  There is uncertainty at each level.  In a perfectly objective world, we'd be able to delve arbitrarily deep into the hierarchy, allowing evidence to determine the posterior distribution of all hyperpriors.  However, for tractability, the model must be truncated

somewhere- to paraphrase Newton, we must let ourselves stand on the shoulders of giants/

9. I think the only example of "belief" in this paper is where an "expert" persists in making a
judgement despite evidence to the contrary. I think this is rare – usually there is objective
evidence for a judgement, even though the evidence is incomplete. A good scientist recognises that it is incomplete and is open to new evidence.
More generally, a good scientist holds his/her views tentatively and hypothetically, recalling that scientific progress takes place through the replacement of one false hypothesis by a better (but probably false) hypothesis. Consequently, a good scientist tries not to "believe" anything but to work via a series of hypotheses and assumptions and their testing.

Not all scientists will be familiar with all of the evidence, and some may be over-confident. Part of the motivation here is to find a way of combining the expert judgements of many scientists, relying on the observation that the scientists who are not open to new evidence are a minority group, and so will only have a small impact on the end consensus result.

Detailed comments

11. l.7. Here and many other places. "beliefs". See General Comments above.
We've replaced "beliefs" in most places

12. l.9. Here and many other places. "subjective". See General Comments above.
For the reasons above, we've kept "subjective". We must decide what evidence to use, assess its quality, choose a model (or candidate models) to interpret it, specify priors on model parameters, and decide how different lines of evidence relate to one another. All of these are decisions that must be made, and therefore we feel "subjective" is appropriate.

13. l.19, eq.(1). What is $M_0$ ? - the climate system, a model of the climate system, or the
simple energy balance model? If the last, then is the RHS of (1), i.e. including $\Delta N$, different?

We've more clearly defined M0

14. l.33: "aerosols". Net cooling in response to aerosols?

Reworded to "We also have the evidence of the planet itself, which has been steadily warming in response to net anthropogenic forcing, which includes not just emissions of $\rm{CO}_2$ but of other greenhouse gases and aerosols as well."

15. l.49 and l.170: "knowledge". See General Comments above.
We've removed most references to "belief"

16. l.61-622, eq.(3) and following line. If P(Θ) is a belief, then P(Y|Θ) must also be a belief
(a posterior belief). See General Comment 7 above.
This section has been rewritten

17. Fig.1(a). Axes need labels.

Fixed.

18. l..120, equation. This is not very clear. C is not defined.
This section has been rewritten to emphasize the "probability mass" concept- please see response to R1 above.

19.
l.161-162. Sentence "These incorporate expert judgement ...". These are normally objective, not subjective, i.e. they are inter-subjectively shared and criticised. This is fundamental in science.

Yes, expert judgment should be shared and criticized (and this is one of the goals of this paper) but this does not necessarily make such judgements, especially about an uncertain quantity, "objective"- merely that frameworks such as this one that seek to interrogate and synthesize these judgments using evidence are necessary.

20.
l.166-167: "well-informed scientist". Again, informed by objective information.21.

Unfortunately, well-informed scientists may still have imperfect knowledge or different opinions on the literature

l.217-218: "Why do these two distributions not overlap substantially?" They appear to overlap substantially - they are well within each other's one-sigma points.
This is fair, and we've removed this. We're only trying to illustrate that the two distributions overlap more if a model with state dependence in the LGM is used.
.

22.
l.229: "odds". This is another word associated with a subjective theory of probability,

and best avoided if you adopt an objective approach.

This is likely another example of where we disagree due to hierarchy truncation. Yes, an expert should allow evidence to inform her/his judgment of model odds. But that would require "going another step down" in the hierarchy and having a scientific debate over the prior odds, which would in turn require a debate over the evidence informing those prior odds, and so on and so on. We feel it's better, in a tractable analysis, to simply clearly specify these priors.

23.
l.237: "definite". What does definite mean here? Does it mean "certain"? If so, this would not be a scientific statement - uncertainty is all-pervasive in science. If you remove
"definite" from this sentence, do you not conclude that state dependence is likely?

We've rewritten this section and it no longer appears.

24.
l.242: "We are not arguing that this is the objectively "correct" way to combine the Last Glacial Maximum reconstructions with historical observations." Given my comment above, it is not clear what you are arguing here.

We've rewritten this section and it no longer appears.

25.
l.250: "relying on a community of experts". Yes! - this makes it objective - this is how we do science - inter-subjectively shareable and criticisable.

This is how science *should* be done, but it is not how it *is* done- at least on timescales necessary for publishing assessments! Experts are often unable to reach consensus decisions, and thus a framework that incorporates potentially subjective prior information is necessary. We feel that making decisions transparent is the first step toward such important criticism.

26.
l.263: "assume". Yes - so these are hypotheses (to be tested), not beliefs.
Yes, as we point out in the previous section, the Bayesian model evidence allows us to assess models in light of the evidence. Our ability to reject or accept a hypothesis, however, depends strongly on the prior odds, and is - in a truncated model hierarchy- subjective.

27.
l.286: "prior assumptions: Yes - much better! You can assume something without believing it.
Ok, we've kept this

28.
l.290: "accurate". Meaning exact? Unusually, accuracy means a quantification of the Uncertainty.

Replaced with "Similarly, we might set the prior on $\mu$ using the result of a single published study (say, for example, $\Delta T$ from T20). "

29.
l.294: "belief". At no point in the discussion contained in this paragraph do you need to "believe" anything - you are making certain assumptions or posing certain hypotheses, and then
testing their consequences.

We've removed "belief"

30.
l.337: "the prior beliefs of two hypothetical experts". Or you could say just two hypotheses?
31.
l.341: "However, some experts may not be so open-minded ...". So, are you are saying that there are closed-minded experts who "believe" things and open-minded experts who make hypotheses?

Perhaps we're saying that there are some experts who might be considered by some to be unscientific. But we need to include their views to give a consensus estimate. Also it's not a clear-cut thing - there's a sliding scale depending on how narrow their priors are.

32.
l.343-345: "Expert C's confidence remains unshaken ..." and following sentence.
This is fundamental to how science works. You are saying that Expert C is not influenced by
evidence and so is not behaving rationally/scientifically. In (good) science, we suspend belief and act tentatively and hypothetically.

Yes, the idea is that this method allows you to include the views of all experts, whether they are willing to modify their views or not. This is what IPCC does, and we're trying to do the same, but in a more transparent way. Hopefully their contributions will be downweighted but not entirely ignored. This avoids you having to screen out people who you think might be overconfident, something that can be problematic, when building a community assessment.

33.
l.348-349: "The narrowness of C's prior ...". It's OK to have a narrow prior, if all the evidence you have (at present) points in that direction, but it is prudent to assume that there is some possibility (low probability) of a gross error, because of some effect that

[revised manuscript text omitted]

 When combined with a uniform prior on $\lambda$, the model evidence for $M_\alpha$ is the area under the green curve in Figure 5b.

Using S20 evidence and these priors, we find that the Bayes Factor is 1.33. This means that if our prior is that both models are equally likely, the evidence shifts those odds: the model depicted in panel  b is about 33% more likely to have generated

380 the observed paleo and historical evidence.

 However, using T20 evidence, the Bayes factor is 0.93. This suggests that the "better" model to use, given T20 evidence, is one without state dependence

385 . Clearly, the "best"

 model depends on the evidence used, the prior knowledge of whether we are comparing "apples to apples"

390 ~~evidence and assuming no state dependence (orange line, Figure 5(c)) closely overlaps the historical likelihood, as does the likelihood assuming state dependence with a prior on $\alpha$ as in S20 (red line, Figure 5(c)). The latter model, however, yields a broader likelihood for $\lambda$ and therefore the region of overlap with the historical evidence is smaller. The Bayes factor using T20 evidence is calculated as~~

$$BF = \frac{P(Y|\mathcal{M}_{\alpha,\Delta\lambda})}{P(Y|\mathcal{M}_{0,\Delta\lambda}} = 0.93.$$

395 , and the

**7**

 priors we place on $\lambda$ , $\Delta\lambda$, and $\alpha$.

We note that whether the twin peaks problem is indeed a "problem" is largely dependent on the prior odds $P(M_\alpha)/P(M_0)$,

400 which must be specified. If we have prior knowlege that the  two lines of evidence are measuring the same thing, then we will give more prior weight to the simple model $M_0$ and the Bayes Factor will do little to shift the odds. This will result in a narrower posterior estimate: if two lines of evidence

405  are compatible only for a small range of values, and we are confident in what the evidence is telling us, then we may be more confident in its posterior value.

**7 A Way Forward**

Thus far, we have established that there are three places where unavoidable subjective decisions must be made: collecting evidence, choosing the interpretive model, and assessing prior knowledge of that model's parameters. We have also established that multiple lines of evidence  appear more or less compatible depending on the models used. Here, we present a suggested  framework for making these decisions in a community assessment framework.

**7.1**

**7.1 Handling evidence uncertainty**

 Whether and how much a newly published estimate of a particular quantity (for example, $\Delta T$ or $\Delta F$ from the Last Glacial Maximum) affects the evidence base depends on prior knowledge of that quantity. It also depends on expert assessment of how the new study relates to existing literature. A single highly certain, high-quality study can strongly shift previously uncertain estimates, while low-quality or uncertain published estimates may not change previously firm understandings.

We suggest formalizing these intuitions using a Bayesian random effects meta-analysis  Smith et al. (1995) , frequently used in fields as diverse as psychology  Gronau et al. (2021) , medicine Sutton and Abrams (2001), and ecology  Koricheva et al. (2013) . This model can be written as

$$\underset{\sim}{\underline{y_i}\hat{y}_j} \sim \underline{\mathcal{N}}\underset{\sim}{N}(\underline{\theta_i y_j}, \sigma_{\underline{i}j}) \tag{9}$$

$$\underline{\theta_i}y_j \sim \underline{\mathcal{N}}\underset{\sim}{N}(\underline{\mu}\underset{\sim}{Y}, \tau)\underline{\mu \sim g(.)\tau \sim h(.)} \tag{10}$$

 where $\hat{y}_j$ and $\sigma_j$ are the reported mean and  standard deviation of each study $j$. We assume the true (latent)  mean $y_j$ of each study is normally distributed about an overall mean $Y$, with $\tau$

The priors  the expected inter-study standard deviation.

440    The priors we put on the quantities of interest– the overall mean $Y$ and the between-study spread $\tau$ any differences are simply due to sampling error– quantify our previous knowledge of and views about the literature. A $\tau$ very close to zero suggests homogeneity across studies (and, in fact, choosing to set $\tau = 0$ reduces the random-effects model to the fixed-effects model). By contrast,

445     if we have reason to believe that multiple studies should vary in their reported values due to structural and design factors, then we might place a broad prior on $\tau$. For example, a fixed-effects model might be appropriate for calculating the ensemble mean of a quantity within a single CMIP model, whereas a random-effects model might be more appropriate for combining ensembles of multiple CMIP models, which we know to differ structurally.

As  a specific example relevant to calculating the feedback parameter $\lambda$,  consider multiple published LGM global

450    mean temperature changes $\Delta T$ derived from proxies and models as well as from PMIP3 and PMIP4 models (Table 1).

~~How best should we combine these estimates to obtain a single distribution of $\Delta T$ and its uncertainty? This depends on many things: the literature on which we rely, our judgement about how best to pool multiple estimates into a single uncertain quantity, and the reported quantities in the studies themselves. How cold was the Last Glacial Maximum? The answer depends on your prior beliefs about the cooling and about the literature. Shown are posterior distributions for the LGM cooling $\Delta T$~~

455

Figure 6 illustrates how the posterior distribution of $\Delta T$ depends on prior beliefs about the nature and quality of the published literature assessing it. Consider, for example, a random-effects model in which we place broad priors on the mean $\mu \sim N(0, 100)$ and inter-study standard deviation $\tau \sim U(0, 100)$. With these prior assumptions, 90% of the resulting posterior

460    density for $\mu$ (the true value of $\Delta T$) lies between (-5.9K, -4.8K). Assuming that there is *no* inter-study spread (i.e, $\tau$ is assumed to be zero with zero uncertainty: a fixed effect model) would yield an estimate of $\Delta T$ 90% likely to be between -4.8 and -4.5K. This much narrower (and warmer) estimate results from the extremely restrictive prior belief that every study, regardless of method, targets the same underlying $\Delta T$ and would yield the same results if performed perfectly and with adequate data. Similarly,  we might set the prior on $\mu$ using the result of a single published study (, for example

465     $\Delta T$ from T20). Combined with a broad uniform prior on the inter-study spread, this results in an 90% posterior density estimate of (-6.2K, -5.6K). If, however, we adopt the restrictive fixed effects model, the T20 study is merely treated as an outlier and fails to substantially move the posterior distribution toward cooler values of $\Delta T$ (red line), even

[revised manuscript text omitted]

von Deimling, T. S., Ganopolski, A., Held, H., and Rahmstorf, S.: How cold was the Last Glacial Maximum?, Geophysical Research Letters, 33, https://doi.org/10.1029/2006gl026484, 2006.

Winton, M., Takahashi, K., and Held, I. M.: Importance of ocean heat uptake efficacy to transient climate change, Journal of Climate, 23, 2333–2344, 2010.

Yue, X., Wang, H., Liao, H., and Jiang, D.: Simulation of the direct radiative effect of mineral dust aerosol on the climate at the Last Glacial Maximum, Journal of Climate, 24, 843–858, 2011.

Zhu, J. and Poulsen, C. J.: Last Glacial Maximum (LGM) climate forcing and ocean dynamical feedback and their implications for estimating climate sensitivity, Climate of the Past, 17, 253–267, 2021.

has not been handled correctly. This leads to a different pdf (e.g. Gaussian + constant). This is a common problem is science - it's called quality control.

Perhaps our approach can be seen as a quality control method?  If one expert's prior is much narrower than others, assuming all have access to the same knowledge, then that suggests the first expert is being overconfident.

34.
l.379: "The beauty of Bayesian methods ...". The beauty of objective Bayesian methods is that you don't need to deal in "belief" at all.
We have replaced "beliefs" with "knowledge"

35.
l.396, eq.(A3). Superficially, there appears to be a minus sign missing here – required for
a Gaussian shape.
The integral is correct- it's a commonly used form in quantum field theory.  Note that this is the line integral over the curve defined by the model (and that lambda is negative in B).

36.
Equations at the end of section A1. Again, not clear how this leads to a Gaussian shape.
The integrals are correct

Editorial Comments all fixed

37. l.242: typo "is are".

38. l.260, 261.: typos "s(S", "y(K"

39. A2, line 1: typo "this is method is distinct".